# Spatial Discriminability of CLIP for Training-Free Open-Vocabulary Semantic Segmentation

## Abstract

Extending CLIP models to semantic segmentation remains a considerable challenge, largely due to the misalignment between their image-level pre-training objectives and the pixel-level spatial understanding required for dense predictions. Prior efforts have achieved encouraging results by reorganizing the final layer and feature representations of CLIP to enhance dense predictions. However, these approaches often inherit the global alignment bias of the final layer, leading to suboptimal spatial discriminability and segmentation performance. In this work, we propose TLH-CLIP, a novel training-free framework that systematically exploits the spatial discriminability across *Token*, *Layer* and *Head* levels in CLIP for dense predictions. Through comprehensive analysis, we uncover three key findings: (i) some anomalous tokens emerges in the final layers, which are category-agnostic but disproportionately attract attention from semantically meaningful patch tokens, thereby degrading spatial discriminability; (ii) the final few layers primarily enhance global image-text alignment with great sacrifice of local discriminability (e.g., last 3 layers in ViT-B-16 and 5 layers in ViT-L-14); (iii) a few attention heads (e.g., 10 out of 144 in ViT-B/16) demonstrate strong spatial discriminability across different datasets. Motivated by these insights, we propose three complementary techniques: abnormal token replacement, semantic-spatial reweighting, and selective head enhancement to effectively recover spatial coherence and improve segmentation performance without any additional training, auxiliary pre-trained networks, or extensive hyperparameter tuning. Extensive experiments on 8 common semantic segmentation benchmarks demonstrate that TLH-CLIP achieves state-of-the-art performance across diverse scenarios, highlighting its effectiveness and practicality for real-world deployment.

## 1 Introduction

Recent advances in vision-language pretrained models, such as CLIP [1], have demonstrated remarkable generalization and open-vocabulary recognition capabilities at the image level, thereby opening up possibilities for transferring image-text alignment to pixel-level tasks. Despite this progress, they often underperform in dense prediction tasks like semantic segmentation, primarily due to their limited capacity to localize fine-grained visual details [2, 3]. To address these limitations, several studies have incorporated trainable modules into CLIP, typically relying on additional forms of supervision such as dense annotations for a restricted set of categories [4, 5, 6, 7] or supplementary image-text pairs [8, 9, 10, 11, 12]. Although these approaches have demonstrated improved segmentation performance, they incur significant computational and annotation costs. Furthermore, the dependence on limited supervision often undermines the generalizability of the model, making it prone to overfitting the training distribution.

These challenges have sparked increasing interest in training-free methods[3, 13, 14, 15, 16, 17, 18, 19, 20], which aim to adapt CLIP's pre-trained representations for semantic segmentation without additional training, while preserving its generalization capability. A key difficulty in this direction is enhancing spatial representations for accurate pixel-level predictions. For instance, MaskCLIP[14] computes similarity between key features in the final attention layer to enrich patch embeddings. SCLIP [3] replaces the standard query-key attention with correlative self-attention (query-query and key-key). ClearCLIP [15] further removes residual connections and discards the FFN in the final layer to reduce noise and improve spatial alignment. ResCLIP [20] incorporates attention maps from earlier layers to refine final-layer attention map. However, these methods largely focus on modifying the final-layer attention, often leading to suboptimal ambiguous local relationships and noisy segmentation. To address spatial limitations, some approaches incorporate features from auxiliary backbones such as DINO [21, 17], SAM [17, 22], or diffusion models [23, 24]. While effective, these methods incur significant computational and memory overhead.

Motivated by these limitations, we begin with a layer-wise analysis of spatial discriminability and text-semantic alignment within the CLIP model. As shown in Figure 1, we observe a clear spatial-semantic trade-off in the final layers: spatial discriminability drops sharply, while the improvement in semantic alignment is relatively marginal. To understand the cause of this phenomenon, we further examine internal token interactions and structural patterns across layers. Through attention map visualizations, we find that certain abnormal tokens emerge in the deeper layers, attracting disproportionately high attention from nearly all spatial positions. This behavior causes the majority of tokens to converge on a small subset, thereby disrupting the spatial coherence of the representation. Further analysis reveals that these abnormal tokens exhibit sparse and high-magnitude activations. Moreover, they are class-agnostic, as their similarity remains consistent across different positions, layers, and input samples, indicating a lack of semantic specificity. Contrary to prior assumptions that such tokens encode global semantic content, our findings suggest they may instead function as bias components that offset global-mean features, thereby facilitating alignment with text embeddings.

Based on the analysis, we propose TLH-CLIP, a training-free framework that leverages the inherent properties of CLIP to enhance the spatial discriminability of visual features while preserving their semantic alignment. TLH-CLIP comprises three complementary strategies: abnormal token replacement (ATR), spatial-semantic reweighting (SSR), and selective head enhancement (SHE). Specifically, the ATR employs hoyer scores to identify abnormal tokens by thresholding their characteristic sparsity. Once detected, these anomalous tokens are replaced with a weighted average of normal tokens, based on spatial distance. To mitigate the degradation of spatial discriminability in the earlier final layers, SSR reweights the contributions of the residual pathway relative to the attention and FFN submodules. This adjustment restores a better balance between spatial coherence and semantic abstraction, leveraging the fact that late-intermediate layers exhibit stronger spatial discriminability while maintaining comparable levels of semantic alignment. Finally, SHE further enhances spatial coherence by selectively aggregating features from attention heads with high spatial discriminability, using them to refine the output representations. Experimental results demonstrate that TLH-CLIP achieves significant performance improvements when integrated into various baseline methods, establishing new state-of-the-art results across eight benchmark datasets.

**Contributions.** Our contributions can be summarized as follows:

• We conduct a comprehensive analysis of spatial discriminability at the token, head, and layer levels.

• We propose TLH-CLIP, a novel training-free approach, terms TLH-CLIP. To the best of our knowledge, this is the first work to explicitly modify the inference procedure prior to the final layer, enabling improved spatial coherence without compromising semantic alignment.

• The extensive experiment results on open-vocabulary semantic segmentation tasks consistently demonstrate the effectiveness of the proposed method.

## 2 Analysis

### 2.1 Preliminaries

CLIP employs a Vision Transformer (ViT) [25] as its image encoder to generate visual representations that are aligned with corresponding textual descriptions. The vision encoder first tokenizes an input image of size $H \times W \times 3$ by dividing it into a grid of non-overlapping patches of size $P \times P$,

yielding $h = H/P$ rows and $w = W/P$ columns of patches. Each patch is then linearly projected into a $D$-dimensional embedding space, $\mathbf{x}_i \in \mathbb{R}^D$, and augmented with positional embeddings. An additional learnable [CLS] token is prepended to the sequence and is later used for image-level prediction. The resulting token sequence is denoted as $\mathbf{X}^0 = [\mathbf{x}_{\text{cls}}^0, \mathbf{x}_1^0, \ldots, \mathbf{x}_{hw}^0] \in \mathbb{R}^{(1+hw) \times D}$. This sequence is passed through a stack of $L$ Transformer encoder layers, each consisting of a multi-head self-attention (MSA) module followed by a feed-forward network (FFN). Let LN denotes layer normalization, the token representations are updated at each layer $l$ as follow:

$$\hat{\mathbf{X}}^l = \mathbf{X}^{l-1} + \text{Attn}(\text{LN}(\mathbf{X}^{l-1})), \tag{1}$$

$$\mathbf{X}^l = \hat{\mathbf{X}}^l + \text{FFN}(\text{LN}(\hat{\mathbf{X}}^l)). \tag{2}$$

The CLIP model is originally trained on large-scale image–text pairs for open-vocabulary image recognition tasks. To extend it to semantic segmentation, a natural approach is to compute the similarity between the visual tokens $\mathbf{X}^L = [\mathbf{x}_1^L, \ldots, \mathbf{x}_{hw}^L]$ from the final Transformer layer and the textual embeddings of $C$ category names, denoted by $\mathbf{t} \in \mathbb{R}^{C \times D}$. This results in a patch-text similarity map of size $hw \times C$. Denote $\mathbf{t}_c$ as the embedding of the $c$-th class name, the final segmentation prediction is obtained by applying an `argmax` operation over the class dimension of this similarity map, as follows:

$$\hat{c}(\mathbf{x}_i) = \arg\max_c \frac{\langle \mathbf{x}_i^L, \mathbf{t}_c \rangle}{\|\mathbf{x}_i^L\| \cdot \|\mathbf{t}_c\|}, \tag{3}$$

Ideally, for effective semantic segmentation, the vision encoder should produce feature representations that satisfy two key properties:

- **Spatial discriminability (SD)**: token features should exhibit high internal consistency within the same semantic category while remaining clearly distinguishable from those of other categories, thereby enabling accurate and clean segmentation results.

- **Semantic alignment (SA)**: token features should be well-aligned with their corresponding textual embeddings to enable semantically meaningful segmentation results.

Beyond their importance in open-vocabulary semantic segmentation, these two properties are also more highly relevant to the development of multimodal large language models (MLLMs), as the vision encoder of CLIP is often directly employed to extract visual representations without additional training, serving as input to downstream language models such as LLaVA [26, 27]. In this work, we aim to enhance the spatial discriminability of CLIP features in a training-free manner, thereby preserving the its strong generalization capability.

## 2.2 Analysis of layer-wise spatial discriminability and semantic alignment

**Significant decline in SD with marginal gains in SA in the final layers.** To assess whether CLIP visual features exhibit the desired properties, we investigate the layer-wise SD and SA within CLIP models. To quantitatively assess SD property, we follow the evaluation protocol proposed in [28]. In particular, we extract patch-level feature representations from the vision encoder for each image and associate them with corresponding semantic labels using the ground-truth segmentation masks from Pascal VOC [29], PASCAL Context [30], ADE20K [31], and COCO-Stuff [32] datasets. Specifically, let $\mathbf{x}_i^l \in \mathbb{R}^D$ and $\mathbf{x}_j^l \in \mathbb{R}^D$ denote the feature representations of two image patches $i$ and $j$ extracted from the $l$-th layer of the encoder. These feature vectors are $\ell_2$-normalized, and their cosine similarity is computed to serve as the prediction of a binary classifier that indicates whether the two patches belong to the same semantic category. Given the corresponding semantic labels $t(\mathbf{x}_i)$ and $t(\mathbf{x}_j)$, the target value for classification is set to 1 if $t(\mathbf{x}_i) = t(\mathbf{x}_j)$, and 0 otherwise. To evaluate the SA property, we extract the intermediate representations $\mathbf{x}_i^l \in \mathbb{R}^D$ from each individual visual token at layer $l$, and use them as inputs to the final layer to project these features into the final visual latent space for semantic prediction. Following [15], we remove the FFN and residual connections in the final layer to avoid introducing contaminating semantic information. Additionally, inspired by [14], we replace the last-layer attention matrix with an identity matrix to avoid noisy integration during the final attention computation. The final visual representation of each layers can be expressed as $\mathbf{v}_i^l = \mathbf{x}_i^l \mathbf{W}_v^L \mathbf{W}_o^L \in \mathbb{R}^D$, where $\mathbf{W}_v^L$ and $\mathbf{W}_o^L$ denotes the value and output project matrix in last-layer MSA module. Based on these representations, SA is measured using the average accuracy between the predicted and ground-truth semantic labels, following Equation (3).

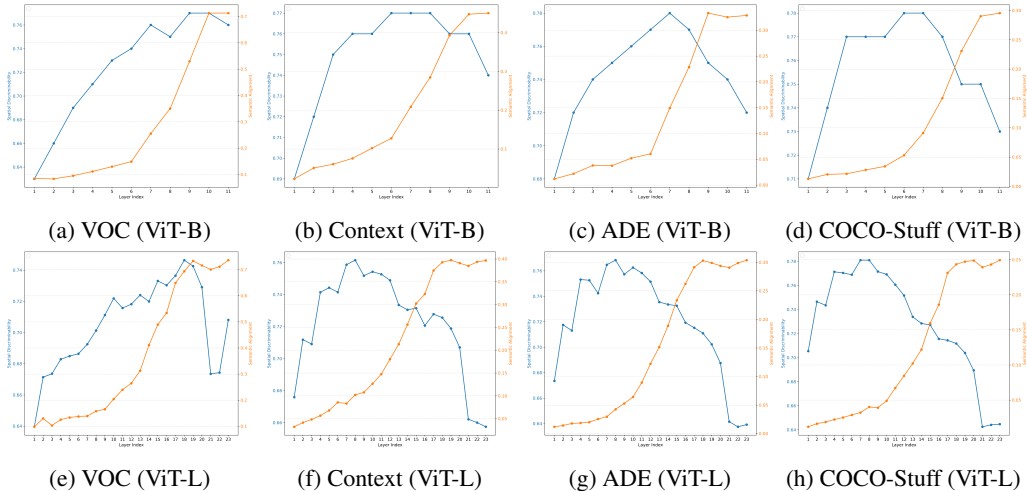

Figure 1: Layer-wise analysis of spatial discriminability (blue curves) and semantic alignment (orange curves) within the CLIP vision encoders across different datasets. The final layer is excluded from the analysis to avoid discrepancies caused by prior modifications to the last-layer in different methods.

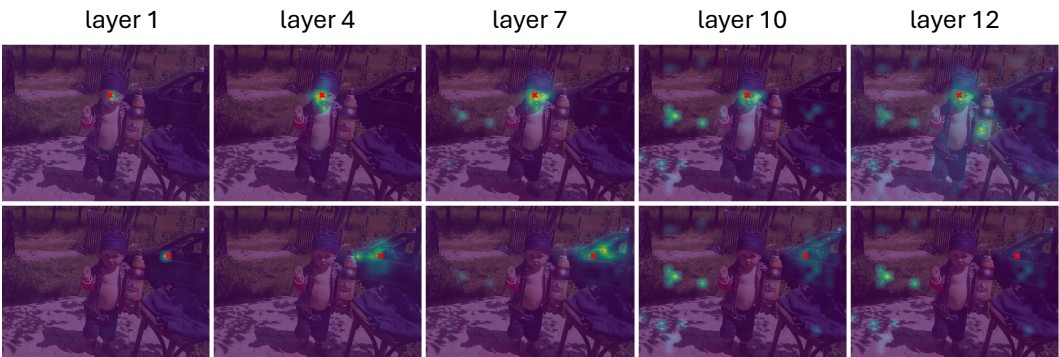

Figure 2: Visualization of the abnormal token phenomenon in the attention maps across different layers of the ViT-B/16 model in the CLIP vision encoder.

We present the layerwise SD and SA scores for both the ViT-B/16 and ViT-L/14 models used as the CLIP vision encoder in Figure 1. From the results, we make the following observations:

- The SD of CLIP exhibits an inverted U-shaped curve across layers: it initially increases in the early stages but declines in the deeper layers. This decline is especially prominent in the final layers. For example, the last two layers ( (excluding the final layer)) of the ViT-B/16 model and the last five layers of the ViT-L/14 model show a marked reduction in spatial discriminability.

- SA follows an approximately monotonic increasing pattern across layers: it improves substantially in the early layers but gradually saturates in the final layers, offering only marginal gains thereafter.

These findings offer a nuanced understanding of why CLIP has proven effective for open-vocabulary semantic segmentation. In particular, the strong semantic alignment observed in the final layers explains why prior work often leverages last-layer features for aligning visual tokens with textual categories. However, the significant decline in spatial discriminability in these layers reveals a key limitation as they may lack the fine-grained spatial distinctions necessary for producing accurate and precise segmentation masks. In this work, we aim to address this limitation by proposing methods that jointly preserve spatial structure and semantic alignment through a systematic exploitation of spatial discriminability across token, layer, and head levels. Before introducing our approach, we first investigate the underlying causes of the decline in spatial discriminability in the next subsection.

## 2.3 Analysis of abnormal tokens

**Class-agnostic sparse and large-norm tokens.** To understand the progression within the vision encoder, we analyze attention maps across layers. As shown in Figure 2, deeper layers exhibit a small set of dominant tokens that receive disproportionately high attention from nearly all spatial locations, causing most tokens to focus on this subset, consistent with prior observations [33, 18].This leads to a gradual decline in spatial discriminability, which is essential for accurate segmentation. To further characterize these dominant tokens, we compare their features with those of normal tokens. As illustrated in Figure 3, dominant tokens exhibit sparse and consistent activation patterns, with only a few channels maintaining high activation. To quantify this sparsity, we adopt the hoyer score [34]:

$$\mathcal{H}(\mathbf{x}_i^l) = \frac{\sqrt{D} - \frac{|\mathbf{x}_i^l|_1}{|\mathbf{x}_i^l|_2}}{\sqrt{D} - 1}, \tag{4}$$

where $\mathbf{x}_i^l \in \mathbb{R}^D$ is the feature vector of the $i$-th token at layer $l$. We use this metric to quantify sparsity and visualize its distribution across layers and token positions in Figure 3(b). To evaluate whether dominant tokens encode meaningful semantics, we analyze their pairwise cosine similarity across spatial locations, layers, and image samples on the ImageNet validation set. As shown in Figure 4, these tokens exhibit strong invariance across positions and inputs, indicating limited semantic specificity. Contrary to prior assumptions that they capture global semantic content, our results suggest they act more like bias components that offset global-mean features, facilitating text alignment, similar to the bias term in final-layer classifiers under neural collapse [35, 36].

## 3 Method

In this section, we provide a detailed description of our training-free framework, which comprises three components: Abnormal Token Replacement (ATR) in Section 3.1, Spatial-Semantic Reweighting (SSR) in Section 3.2, and Selective Head Enhancement (SHE) in Section 3.3. Each component is complementary, and together they work synergistically to enhance the spatial discriminability of the CLIP model, based on our previous analysis.

### 3.1 Abnormal token replacement (ATR)

To mitigate the adverse effects of these anomalous tokens, we propose a simple yet effective strategy to suppress their influence prior to the final layer. As demonstrated in our earlier analysis, these tokens exhibit characteristically sparse activation patterns. To systematically identify them, we employ the hoyer score $\mathcal{H}(\mathbf{x}_i^l)$ defined before as a sparsity-based criterion. Tokens with scores exceeding a predefined threshold $\tau$ are deemed anomalous and grouped into the set $\mathcal{A}_l = \{i | \mathcal{H}(\mathbf{x}_i^l) > \tau\}$. After identifying them, we suppress their influence using an unnormalized 2-dimensional Gaussian kernel. Specifically, each anomalous token at spatial position $(m, n) \in \mathcal{A}$ is replaced by a weighted aggregation of its neighboring non-anomalous tokens:

$$\mathbf{x}_{m,n}^l = \frac{\sum_{i=1}^{w} \sum_{j=1}^{h} w_{m,n,i,j}^l \mathbf{x}_{i,j}^l}{\sum_{i=1}^{w} \sum_{j=1}^{h} w_{m,n,i,j}^l}, \quad \forall (m, n) \in \mathcal{A} \tag{5}$$

$$w_{m,n,i,j}^l = \begin{cases} 0, & \text{if} \quad (i, j) \in \mathcal{A} \\ \exp\left(-\frac{(m-i)^2 + (n-j)^2}{2\sigma^2}\right), & \text{otherwise} \end{cases} \tag{6}$$

Here, $\sigma$ controls the spatial extent of smoothing, and the weights $w_{m,n,i,j}$ ensure that only normal tokens contribute to the reconstruction of anomalous ones. Empirically, we find that applying this strategy before the penultimate layer leads to a performance drop, likely due to the removal of inherent biases encoded in abnormal tokens, which substantially alters the inference process. Therefore, we apply it only at the penultimate layer, i.e., with $l = L - 1$.

### 3.2 Spatial-semantic reweighting (SSR)

After mitigating the influence of anomalous tokens in the input to the last layer, the model exhibits improved spatial discriminability. However, a critical challenge remains: anomalous tokens present

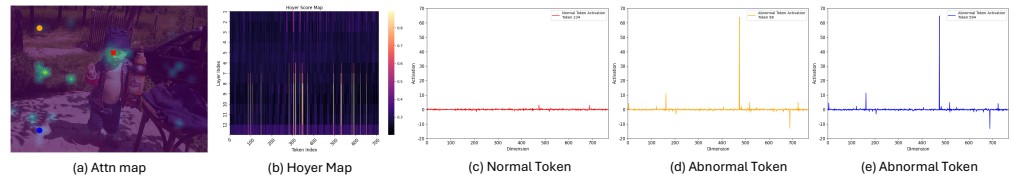

| (a) Attn map | (b) Hoyer Map | (c) Normal Token | (d) Abnormal Token | (e) Abnormal Token |

Figure 3: Illustration of the sparsity and high-norm characteristics of abnormal tokens. Figure (a) shows the attention map of the red anchor token. Figure (b) presents the Hoyer score distribution across layers and spatial positions. Figures (c)–(e) depict the channel activations of a normal token (red) and two abnormal tokens (yellow and blue) highlighted in Figure (a).

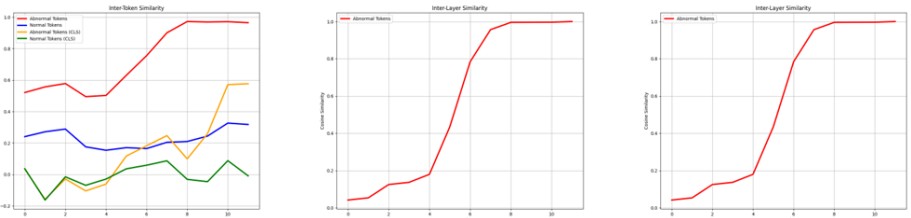

Figure 4: Layer-wise cosine similarity among abnormal tokens across positions, layers and samples.

in earlier layers may have already disrupted the spatial coherence of feature representations, limiting the effectiveness of final-layer refinements. Based on our layer-wise analysis, the final few layers overly emphasize alignment with text embeddings, the marginal gains in semantic alignment come at the cost of a pronounced decline in spatial discriminability. To address this imbalance, we propose a spatial-semantic reweighting strategy that enhances the model's spatial awareness while preserving its semantic alignment capabilities. Given the feature representation $\mathbf{X}^{l-1}$ at the $l$-th layer within the final few layers (e.g., layers 10–11 in ViT-B/16 and layers 20–23 in ViT-L/14), we reweight the forward pass by upweighting the residual pathway and downweighting the attention and MLP submodules, as follows:

$$\hat{\mathbf{X}}^l = (1 + \alpha)\mathbf{X}^{l-1} + (1 - \alpha)\text{Attn}(\text{LN}(\mathbf{X}^{l-1})), \tag{7}$$

$$\mathbf{X}^l = (1 + \alpha)\hat{\mathbf{X}}^l + (1 - \alpha)\text{FFN}(\text{LN}(\hat{\mathbf{X}}^l)), \tag{8}$$

where $\alpha \in [0, 1]$ is a reweighting coefficient that controls the relative degree of emphasis on the residual signal. As $\alpha$ increases, the $l$-th block increasingly preserves spatially discriminative features from earlier layers via the residual pathway, while diminishing the dominant influence of semantic aggregation in the attention and MLP submodules. To the best of our knowledge, prior work has primarily focused on reforming the final layer or modifying its representations to improve performance. However, these approaches often inherit the global semantic alignment bias inherent in the final few layers, resulting in a substantial decline in the spatial discriminability of the extracted features. In contrast, our SSR strategy explicitly mitigates this limitation by rebalancing the contributions of residual and semantic components in intermediate layers preceding the final layer.

### 3.3 Selective head enhancement (SHE)

**Strong spatial discriminability of some attention heads.** While the proposed strategies effectively enhance the spatial discriminability in the final layers, the overall spatial discriminability of the features output by the CLIP vision encoder may still remain suboptimal. Inspired by recent studies [37, 38] revealing that different attention heads capture distinct visual concepts, such as number, shape and texture, this motivates us to investigate whether certain heads are specifically responsible for encoding spatial discriminability. To identify such heads, we follow the formulation introduced in [39, 37], which rewrites the multi-head self-attention (MSA) output as a summation over $H$ independent attention heads: $\text{Attn}(\text{LN}(\mathbf{X}^l)) = \sum_{h=1}^{H} \mathbf{A}_h^l \mathbf{V}_h^l \mathbf{W}_o^l \in \mathbb{R}^{(1+hw) \times D}$, where $\mathbf{A}_h^l$ and $\mathbf{V}_h^l$ denote the attention and value matrices for the $h$-th head at layer $l$, and $\mathbf{W}_o^l$ is the output projection matrix shared across all heads. We extract the contribution of the $h$-th head at layer $l$ and apply abnormal token resolution as follows:

$$\mathbf{X}^{l,h} = \sigma(\mathbf{A}_h^l \mathbf{V}_h^l \mathbf{W}_o^l), \tag{9}$$

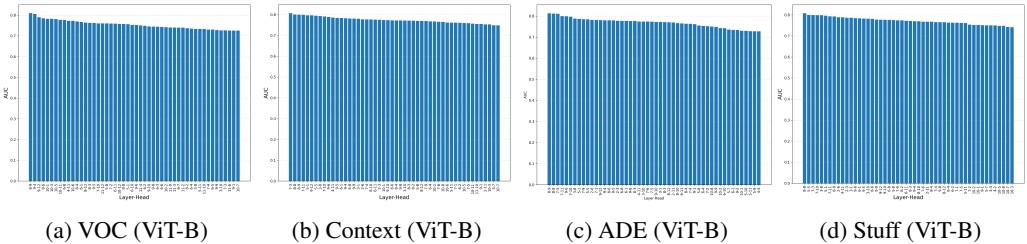

| (a) VOC (ViT-B) | (b) Context (ViT-B) | (c) ADE (ViT-B) | (d) Stuff (ViT-B) |

Figure 5: Head-wise analysis of spatial discriminability within the ViT-B/16 vision backbone across multiple datasets. To ensure consistency, the final layer is excluded, and only the top 50 attention heads are visualized in each figure for clarity.

where $\sigma(\cdot)$ denotes the abnormal token replacement operation defined previously. To assess the SD of each attention head, we adopt the same AUC-based metric as the preceding layer-level analysis. Figure 5 shows the head AUC distribution for ViT-B/16, with ViT-L/14 results in the appendix. From the figure, we observe that the output features from certain attention heads, such as the 9th head in the 8th layer, consistently exhibit high AUC scores across different datasets, suggesting that these heads are more effective at capturing SD information than others.

Building on this observation, we propose to selectively leverage high-performing heads to enhance the spatial discriminability of the output representations. Let $\text{AUC}_{l,h}^s$ denote the AUC score of the representations from the $h$-th head in the $l$-th layer for dataset $s \in \{\text{VOC, Context, ADE, Stuff}\}$. To obtain a dataset-agnostic measure of discriminability, we compute the average AUC score for each head across all datasets, denoted as $\overline{\text{AUC}}_{l,h}$. The distribution of these average scores is provided in the appendix. We rank all heads by their $\overline{\text{AUC}}_{l,h}$ scores and select the top-$k$ to form the set $\mathcal{H}_{\text{top}k}$. The corresponding feature representations are then aggregated as: $\overline{\mathbf{X}}_{\text{top}k} = \frac{1}{k} \sum_{(l,h) \in \mathcal{H}_{\text{top}k}} \mathbf{X}^{l,h}$. This aggregated feature $\overline{\mathbf{X}}_{\text{top}k}$ is used to construct a similarity map $S = \frac{\overline{\mathbf{X}}_{\text{top}k} \overline{\mathbf{X}}_{\text{top}k}^{\top}}{\|\overline{\mathbf{X}}_{\text{top}k}\|^2}$, which captures the pairwise similarity among visual tokens. To mitigate the influence of spurious interactions between tokens from different semantic categories, we apply a thresholding operation with a predefined parameter $\beta$, resulting in the filtered similarity map $S_\beta$, where $S_\beta(i,j) = S(i,j)$ if $S(i,j) \geq \beta$, and $S_\beta(i,j) = 0$ otherwise. The resulting $S_\beta$ is then column-wise normalized, and subsequently used to refine the final-layer features by $\mathbf{X}^{L-1} = \text{Norm}(S_\beta)\mathbf{X}^{L-1}$.

## 4 Experiment Results

**Evaluation datasets.** We follow the standard evaluation protocol from prior works [3, 15, 16] and assess our method on eight widely used semantic segmentation benchmarks. For clarity, we group them into two categories and use abbreviated names throughout the paper. The first category excludes background and includes Pascal VOC [29] (VOC20), Pascal Context [30] (Context59), COCO-Stuff [32] (Stuff), ADE20K [31] (ADE), and Cityscapes [40] (City). The second includes background and consists of VOC21, Context60, and COCO-Object [32] (Object). We use CLIP [1] models with ViT-B/16 and ViT-L/14 backbones via MMSegmentation [41], and report results using the mean Intersection-over-Union (mIoU). All hyperparameters are fixed across datasets without task-specific tuning. Additional implementation details are provided in the appendix.

### 4.1 Comparison with existing methods.

We compare our approach against a comprehensive set of open-vocabulary semantic segmentation (OVSS) methods, including the direct baseline CLIP [1], as well as several state-of-the-art training-free approaches: MaskCLIP [14], CLIPSurgery [13], SCLIP [3], NACLIP [16], ClearCLIP [15], LAVG [42], and ResCLIP [20]. We also include several influential weakly supervised methods, such as GroupViT [5], ReCo [43], and TCL [8]. Unless otherwise specified, all reported results are taken directly from the respective original papers and ResCLIP [20]. As our method is orthogonal to approaches that primarily target improvements in the final-layer attention, we additionally evaluate its effectiveness when integrated with recent state-of-the-art methods that employ specialized attention mechanisms in the last layer, including SCLIP [3], ClearCLIP [15], and ResCLIP [20]. For fair

Table 1: Performance comparison of our approach with other methods on eight semantic segmentation benchmarks following the evaluation protocol in Section 4. Our results are marked in gray.

| Methods | Training | With a background class | | | Without background class | | | | | Avg. |
|---|---|---|---|---|---|---|---|---|---|---|
| | | VOC21 | Context60 | Object | VOC20 | City | Context59 | ADE | Stuff | |
| ReCo [43] | ✓ | 25.1 | 19.9 | 15.7 | 57.7 | 21.1 | 22.3 | 11.2 | 14.8 | 23.5 |
| GroupViT [5] | ✓ | 52.3 | 18.7 | 27.5 | 79.7 | 18.5 | 23.4 | 10.4 | 15.3 | 30.7 |
| TCL [8] | ✓ | 51.2 | 24.3 | 30.4 | 77.5 | 23.1 | 30.3 | 14.9 | 19.6 | 33.9 |
| CLIP [1] | ✗ | 16.2 | 7.7 | 5.5 | 41.8 | 5.5 | 9.2 | 2.1 | 4.4 | 11.6 |
| MaskCLIP [14] | ✗ | 38.8 | 23.6 | 20.6 | 74.9 | 16.4 | 26.4 | 9.8 | 14.8 | 28.2 |
| CLIPSurgery [13] | ✗ | 55.2 | 18.7 | 27.5 | 79.7 | 18.5 | 23.4 | 10.4 | 15.3 | 31.1 |
| LaVG [42] | ✗ | 62.1 | 31.6 | 34.2 | 82.5 | 26.2 | 34.7 | 15.8 | 23.2 | 38.8 |
| NACLIP [16] | ✗ | 58.9 | 32.2 | 33.2 | 79.7 | 35.5 | 35.2 | 17.4 | 23.3 | 39.4 |
| SCLIP [3] | ✗ | 59.7 | 31.7 | 33.5 | 81.5 | 32.3 | 34.5 | 16.5 | 22.7 | 39.1 |
| +TLH-CLIP (ours) | ✗ | 64.8 | 34.8 | 36.6 | 86.3 | 36.1 | 37.6 | 18.0 | 24.9 | 42.4 (+3.3) |
| ClearCLIP [15] | ✗ | 57.0 | 32.2 | 32.5 | 82.3 | 32.8 | 35.8 | 17.3 | 24.0 | 39.2 |
| +TLH-CLIP (ours) | ✗ | 63.9 | 35.2 | 35.6 | 85.7 | 37.8 | 38.8 | 19.2 | 25.8 | 42.7 (+3.5) |
| ResCLIP [20] | ✗ | 60.0 | 32.7 | 34.0 | 85.5 | 35.6 | 35.8 | 17.7 | 23.8 | 40.6 |
| +TLH-CLIP (ours) | ✗ | 63.9 | 35.5 | 35.3 | 86.8 | 38.2 | 38.2 | 19.1 | 25.5 | 42.8 (+2.2) |

comparison, we exclude the *Semantic Feedback Refinement* module in ResCLIP, as it relies on the computationally expensive PAMR [44] post-processing, which is inconsistent with our evaluation setting. For comprehensiveness, results on the ViT-L/14 architecture are provided in the appendix.

In Table 1, we summarize the performance of various open-vocabulary semantic segmentation models on benchmark datasets using the ViT-B/16 backbone. Our proposed TLH-CLIP consistently enhances the performance of state-of-the-art approaches, including SCLIP [3], ClearCLIP [15], and ResCLIP [20]. Notably, when integrated with ResCLIP [20], TLH-CLIP achieves state-of-the-art results, outperforming leading weakly supervised methods. As a plug-and-play solution, TLH-CLIP yields consistent improvements across all datasets compared to the respective baselines, demonstrating its strong generalization capability. We further evaluate performance on the ViT-L/14 backbone. In line with observations from [20], existing methods generally exhibit a performance drop exceeding 2% mIoU when adapting to a different backbone; for instance, ClearCLIP [15] suffers a notable decline of 2.7% mIoU. In contrast, when augmented with TLH-CLIP, this performance degradation is significantly alleviated, highlighting the robustness of our approach. Across both backbones, TLH-CLIP delivers substantial improvements over baseline methods, validating its effectiveness.

## 4.2 Experimental analysis

In this section, we conduct comprehensive ablation studies to validate the effectiveness of our proposed method. We adopt SCLIP [3] as the baseline, which enhances spatial correlation by modifying the attention mechanism in the final layer, replacing the standard $QK^\top$ attention with a combination of $QQ^\top + KK^\top$. In addition, following prior work [15, 20], we remove the residual connections and FFN from the final transformer block to isolate the impact of attention refinement.

**Analysis of the hoyer threshold parameter $\tau$.** Our method relies on hoyer sparsity to identify anomalous tokens, making the sparsity threshold $\tau$ a critical hyperparameter. We conduct a systematic evaluation, as shown in Table 2. At $\tau = 0.2$, many normal tokens are misclassified, leading to excessive smoothing and degraded performance. As $\tau$ increases to 0.4, performance steadily improves, but plateaus between 0.5 and 0.8, with a decline observed beyond this range. The broad stable region indicates a clear sparsity gap between normal and abnormal tokens, highlighting the robustness of ATR to threshold selection. Based on this analysis, we fix $\tau = 0.5$ for all experiments.

**Analysis of spatial-semantic reweighting parameters and number of Layers.** To evaluate the impact of the reweighting strength $\alpha$ and the range of layers involved, from $l_{start}$ to $l_{end}$, we perform a comprehensive sensitivity analysis. The results are summarized in Table 3. We observe that the best performance is obtained when reweighting is applied to layers 10–11 in the ViT-B/16 backbone. This aligns with our earlier findings that these layers experience a marked decline in spatial discriminability while yielding only marginal improvements in semantic alignment. Extending reweighting to include layer 9 results in a slight gain in spatial discriminability but introduces noisy

Table 2: Study of hoyer sparsity threshold $\tau$.

| $\tau$ | C60 | Obj | C59 | City | Avg |
|---|---|---|---|---|---|
| $\tau = 0.2$ | 0.8 | 2.0 | 1.5 | 1.7 | 1.5 |
| $\tau = 0.4$ | 32.8 | 34.0 | 36.6 | 34.7 | 34.5 |
| $\tau = 0.5$ | 32.8 | 34.2 | 36.7 | 34.7 | 34.6 |
| $\tau = 0.8$ | 32.8 | 33.9 | 36.6 | 34.7 | 34.5 |
| $\tau = 0.9$ | 32.8 | 33.9 | 36.6 | 34.3 | 34.4 |
| baseline | 32.4 | 32.9 | 36.0 | 34.3 | 33.9 |

Table 3: Study of $(l_{\text{start}}, l_{\text{end}}, \alpha)$ in SSR module.

| $(l_{\text{start}}, l_{\text{end}}, \alpha)$ | C60 | Obj | C59 | City | Avg |
|---|---|---|---|---|---|
| baseline | 32.4 | 32.9 | 36.0 | 34.3 | 33.9 |
| (9, 11, 0.1) | 32.7 | 32.0 | 36.5 | 36.7 | 34.5 |
| (10, 11, 0.1) | 33.1 | 33.4 | 36.9 | 35.6 | 34.8 |
| (11, 11, 0.1) | 32.7 | 34.1 | 36.4 | 34.9 | 34.5 |
| (10, 11, 0.05) | 32.8 | 33.7 | 36.4 | 35.0 | 34.5 |
| (10, 11, 0.2) | 32.6 | 31.7 | 36.5 | 36.6 | 34.4 |

Table 4: Study of number of selected heads $k$.

| $k$ | C60 | Obj | C59 | City | Avg |
|---|---|---|---|---|---|
| baseline | 32.8 | 34.2 | 36.7 | 34.7 | 34.6 |
| layer($l = 8$) | 33.9 | 37.1 | 37.1 | 35.0 | 35.8 |
| $k = 1$ | 33.4 | 37.1 | 36.6 | 35.4 | 35.3 |
| $k = 10$ | 34.8 | 37.6 | 37.9 | 36.3 | 36.7 |
| $k = 30$ | 34.7 | 37.3 | 37.9 | 36.4 | 36.6 |
| $k = 50$ | 34.7 | 37.3 | 37.8 | 36.3 | 36.5 |

Table 5: Combination of three strategies.

| Methods | Module | | | mIoU | $\Delta$ |
|---|---|---|---|---|---|
| | ATR | SSR | SHE | | |
| baseline | – | – | – | 33.9 | – |
| | ✓ | ✓ | – | 35.3 | +1.4 |
| | ✓ | – | ✓ | 36.7 | +2.8 |
| Ours | ✓ | ✓ | ✓ | **37.4** | **+3.5** |

semantic signals, ultimately leading to a reduction in segmentation performance. In addition, we examine the effect of varying the reweighting threshold parameter $\alpha$. As $\alpha$ increases from 0 to 0.1, performance improves steadily, indicating a beneficial balance between spatial and semantic cues. However, further increasing $\alpha$ leads to a performance drop, as it incorporates more noisy semantic information from earlier layers and significantly perturbs the input distribution of subsequent layers.

**Analysis of the number of selected heads.** We study the effect of varying the number of top-$k$ attention heads selected for enhancement, as shown in Table 4. Empirically, we find that SHE is most effective when combined with ATR; without ATR, the spatially coherent similarity maps can cause normal tokens to be fused with abnormal ones. Therefore, we adopt the baseline SCLIP model equipped with ATR as our baseline. On the ViT-B/16 backbone, increasing $k$ from 1 to 10 improves segmentation accuracy, as aggregating multiple spatially discriminative heads helps suppress spurious correlations. However, performance declines when $k$ becomes too large due to the inclusion of noisy or less informative heads, which introduce undesired cross-category interactions. We also compare head- and layer-level selection (best $l = 8$), finding that head-level selection consistently performs better, as discriminative heads are distributed across layers, while entire-layer selection introduces irrelevant heads and degrades performance.

**Study of each individual components** In the previous parts, we evaluated the effectiveness of each individual component. Table 5 presents their combinations, which yield a substantial improvement of *3.5* mIoU, achieving a final mIoU of *37.5* on these four datasets. These results highlight the complementary contributions of each module to the overall segmentation performance.

# 5 Conclusion

In this paper, we present a comprehensive analysis of the spatial discriminability of pretrained CLIP models across the token, layer, and head levels. Our study reveals three key findings: (1) the emergence of class-agnostic abnormal tokens with sparse, high-norm activations; (2) a notable decline in spatial discriminability in the final layers, despite marginal gains in semantic alignment; and (3) consistently strong spatial discriminability in specific attention heads. Motivated by these observations, we propose TLH-CLIP, a training-free framework that enhances spatial discriminability while preserving semantic alignment. TLH-CLIP introduces three complementary components: (1) abnormal token replacement, (2) spatial-semantic reweighting, and (3) selective head enhancement. Unlike prior methods that focus on modifying the final attention layer, our approach provides lightweight, plug-and-play modules compatible with existing architectures. Extensive experiments on multiple segmentation benchmarks demonstrate that TLH-CLIP consistently outperforms strong baselines. Moreover, as CLIP vision encoders are often frozen during the training of MLLMs, our findings offer valuable insights for improving visual understanding in broader MLLMs.

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
