# OpenReview forum: "Spatial Discriminability of CLIP for Training-Free Open-Vocabulary Semantic Segmentation"
_NeurIPS.cc/2025/Conference — Submitted to NeurIPS 2025_

### Official Review · Reviewer_mmh1 · 2025-06-12

**Clarity:** 3
**Significance:** 3
**Originality:** 3
**Rating:** 4
**Confidence:** 4

**Summary:**

This paper addresses the challenge of adapting CLIP for semantic segmentation, focusing on the mismatch between CLIP’s image-level training and the pixel-level granularity required for dense predictions. The authors introduce TLH-CLIP, a training-free framework that leverages spatial discriminability at the Token, Layer, and Head levels. They identify three key issues in CLIP’s later layers: the emergence of category-agnostic tokens, reduced spatial discriminability due to global alignment, and a small subset of attention heads with strong spatial sensitivity. Based on these findings, they propose three techniques—abnormal token replacement, semantic-spatial reweighting, and selective head enhancement—to improve segmentation performance. Without retraining or auxiliary models, TLH-CLIP achieves state-of-the-art results on 8 benchmarks, showing its practical value and effectiveness.

**Questions:**

- Have you tried to check how the findings transfer on other vision-text models such as SigLIP?

**Ethical Concerns:**

["NO or VERY MINOR ethics concerns only"]

**Final Justification:**

Technically solid paper with strong empirical results. However, the proposed method relies on manual selection of specific layers and tokens for each model, which limits its scalability. While the approach extends beyond CLIP and is shown to work with other models, it does not offer guidance on how to address the issue during model training, an aspect that would significantly strengthen its impact. I am keeping my score to borderline accept.

**Limitations:**

Yes

**Paper Formatting Concerns:**

No major formatting issue

**Quality:**

3

**Strengths And Weaknesses:**

**Strengths**
- The paper is clearly written and well-structured, making it accessible and easy to follow.
- The work addresses a timely and important issue: the emergence of massive activations in vision transformers. The authors provide meaningful insights into this phenomenon, which could inform future approaches to mitigate its impact.
- The focus on improving patch-level semantic alignment between vision and language in CLIP is particularly valuable, especially in the context of training-free open-vocabulary semantic segmentation.
- The proposed TLH-CLIP framework is thoroughly evaluated, with comprehensive ablation studies that effectively isolate and validate the contributions of each component.
- Experimental results show strong and consistent improvements over existing baselines, highlighting the effectiveness and practical relevance of the proposed approach.

**Weaknesses**
- Line 10: The first claimed finding—“some anomalous tokens emerge in the final layers”—has already been observed in prior work [1], which is also cited in Line 159. This raises questions about whether it constitutes a novel key finding. Given the relevance of [1] to the paper’s core findings, it would be more appropriate to include it in the related work section and clearly distinguish the current work’s contributions from prior observations.
- Since the phenomenon of massive activation has been previously documented, the proposed method appears to function more as a downstream corrective strategy for pre-trained CLIP models rather than a fundamental advance. While effective, the contribution might be seen more as a practical adaptation than a conceptual breakthrough.
- The method is largely empirically driven and deeply tailored to the structure of CLIP, with extensive analysis across tokens, layers, and heads. While this yields strong performance and useful insights, it may limit the generalizability of the approach to other vision-language models such as SigLIP, which may exhibit different architectural behaviors. Expanding the analysis to such models could strengthen the paper’s broader impact.

[1] Darcet, T., Oquab, M., Mairal, J., & Bojanowski, P. Vision Transformers Need Registers. In The Twelfth International Conference on Learning Representations.

---

> ### Author Rebuttal · Authors · 2025-07-30
>
> Thank you very much for your time and valuable feedback on our submission. We appreciate your thoughtful comments and the opportunity to clarify and improve our work. We address your concern in detail below.
>
> **W1**: Line 10: The first claimed finding—“some anomalous tokens emerge in the final layers”—has already been observed in prior work [1], which is also cited in Line 159. This raises questions about whether it constitutes a novel key finding. Given the relevance of [1] to the paper’s core findings, it would be more appropriate to include it in the related work section and clearly distinguish the current work’s contributions from prior observations.
>
> **A1**: We thank the reviewer for highlighting this important point. While prior work [1] has identified that abnormal tokens exhibit large magnitudes and are linearly separable, our findings provide a complementary and deeper characterization of these tokens.
> Specifically, we observe that abnormal tokens not only exhibit high magnitude but also show sparse yet strongly activated neurons. More importantly, these activation patterns are highly consistent across tokens, layers, and images, as evidenced by the cosine similarity values ≥ 0.99 in Figure 4. This consistency suggests that abnormal token representations are dominated by a global bias or mean component, which makes them appear highly similar across instances, irrespective of class. Consequently, these tokens predominantly tend to encode class-agnostic, shared information, potentially overshadowing class-specific features and affecting downstream interpretability. This is in contrast to the previous finding in [1] that studies linear separability of these tokens. The similarity among these abnormal tokens is the main motivation of our abnormal token replacement approach for enhancing spatial discriminability. See Reviewer tN1T A2 for detailed discussion about the difference.
>
> We also appreciate the suggestion to better differentiate our contributions and will revise the manuscript to more clearly emphasize our novel observations regarding activation sparsity, representation similarity, and the global component that underlies abnormal tokens.
>
> **W2**: Since the phenomenon of massive activation has been previously documented, the proposed method appears to function more as a downstream corrective strategy for pre-trained CLIP models rather than a fundamental advance. While effective, the contribution might be seen more as a practical adaptation than a conceptual breakthrough.
>
> **A2**: We thank the reviewer for this thoughtful comment. From our understanding, the reviewer is referring to prior work that has documented the large magnitude norm of abnormal tokens. If we have misunderstood the concern, we would be happy to provide further clarification or engage in additional discussion.
>
> As noted in our response to A1, while previous studies have focused on the high magnitude and linear separability of abnormal tokens, our work provides a complementary and deeper characterization. Specifically, we reveal that these tokens also exhibit activation sparsity, high representation similarity, and are dominated by a global component that makes them highly consistent across tokens, layers, and images. These insights offer a more nuanced understanding of abnormal token behavior and a predominant class-agnostic representation collapse, a phenomenon not emphasized in prior work.
>
> Additionally, we became aware of a contemporary paper [1], which was made publicly available after the NeurIPS submission deadline, and reports related observations of sparse and consistent neuron activations in abnormal tokens. While there is some overlap in findings, our work provides a more comprehensive and systematic analysis. In particular, we explicitly compare abnormal tokens across patches, layers, and images, and show that these tokens exhibit extremely high cosine similarity across the entire ImageNet validation set, reinforcing the idea of dominant global representations.
>
> We will revise the manuscript to clearly distinguish our contributions from prior and concurrent work and to highlight the broader implications of our findings.
>
> **W3 and Q4**. **W3**: The method is largely empirically driven and deeply tailored to the structure of CLIP, with extensive analysis across tokens, layers, and heads. While this yields strong performance and useful insights, it may limit the generalizability of the approach to other vision-language models such as SigLIP, which may exhibit different architectural behaviors. Expanding the analysis to such models could strengthen the paper’s broader impact. **Q4**: Have you tried to check how the findings transfer on other vision-text models such as SigLIP?
>
> **A3 and A4**: We appreciate the reviewer’s thoughtful comment. We believe this concern is closely related to the follow-up question Q4, and we address both together here. If we have misunderstood any part of the question, we would be happy to clarify further.
>
> We agree that evaluating the generalizability of our method beyond CLIP like SigLIP would enhance the broader impact of our work. Since SigLIP shares a similar pretraining paradigm with CLIP, differing primarily in the use of a sigmoid loss function, we expect similar phenomena to emerge, and anticipate that our method would also benefit SigLIP in the OVSS setting. However, our current implementation is based on the NaCLIP codebase using the MMPretrain framework, which does not yet support SigLIP. Due to the time constraints of the rebuttal period, we were unable to complete the necessary integration. We will implement the required components and provide the evaluation results on SigLIP during the discussion phase.
>
> Thank you for your understanding and thoughtful feedback.
>
> [1] Jiang, Nick, et al. "Vision Transformers Don't Need Trained Registers." arXiv preprint arXiv:2506.08010 (2025).

---

> ### Author Response · Authors · 2025-08-04
> **Response to reviewer mmh1**
>
> **A3 and A4**: We thank the reviewer for their patience and thoughtful question regarding the generalizability of our approach beyond the CLIP model. In response to your suggestion, we evaluated our method on the SigLIP model. Specifically, we used the ViT-B variant as the SigLIP vision encoder and assessed our method on five datasets that do not include a background category. The results are shown in the table below:
>
> | Method                | VOC20| City  | Context59 | ADE | Stuff | Avg |
> |----------------------|------|------|-------|------|------|-----------|
> | SigLIP | 48.0 | 20.5 | 18.6  | 11.5 | 12.1 | 22.1     |
> | SigLIP+ClearCLIP         | 5.7 | 2.9 | 1.7  | 0.6 | 1.6 | 2.3      |
> | **SigLIP+ours**             | **59.1** | **23.2** | **23.1** | **13.8** | **15.3** | **26.9**  |
>
> Similar to the CLIP model, the vanilla SigLIP model performs poorly on these datasets, achieving an average performance of only 22.1. Empirically, we observed that prior works such as ClearCLIP—which remove the feed-forward network (FFN) and residual connections and modify the attention mechanism in the final transformer layer—perform significantly worse than the vanilla SigLIP model. We hypothesize that this degradation stems from the architectural difference between CLIP and SigLIP: specifically, the replacement of the linear projector in CLIP with the AttentionPool projector in SigLIP. Modifying the final transformer layer, as done in ClearCLIP, may significantly alter the input distribution to the AttentionPool projector, negatively impacting performance. This warrants further investigation in future work. Accordingly, we adopt the vanilla SigLIP model as our baseline.
>
> Orthogonal to previous work that focuses exclusively on modifications to the final transformer layer, our approach is **grounded in a detailed analysis of spatial discriminability and semantic alignment across the token, head, and layer levels**. Rather than restricting changes to the final stage, we introduce improvements at earlier layers of the model, resulting in more effective and generalizable inference. As shown in the table, our method achieves a substantial improvement of **4.8** points in average segmentation performance, increasing from 22.1 to **26.9**.
>
> It is worth noting that, due to time constraints during the discussion period, we did not conduct extensive hyperparameter ablation or tuning. These results therefore demonstrate not only the **generalizability of our method across different CLIP variants** but also its **robustness to parameter selection**. We will include this discussion in the appendix in the revised version.

---

> > ### Comment · Reviewer_mmh1 · 2025-08-04
> >
> > I appreciate the additional material provided in the rebuttal and acknowledge the authors’ efforts in addressing the reviewers’ concerns. I have decided to maintain my score, mainly for the following reasons. The paper demonstrates strong empirical performance and provides valuable insights into the causes of limited spatial discriminability and semantic alignment in CLIP models. However, the proposed method relies on the manual selection of specific layers and tokens for each model, which limits its scalability. While the authors show that the approach works effectively with other models such as SigLIP, no generalizable rule emerges that could offer guidance on how to address the issue during model training.

---

### Official Review · Reviewer_DEqc · 2025-06-25

**Clarity:** 2
**Significance:** 3
**Originality:** 3
**Rating:** 4
**Confidence:** 3

**Summary:**

The paper presents a training-free framework for improving CLIP-based open-vocabulary semantic segmentation models. Through comprehensive layer-wise analysis, the authors demonstrate that CLIP's semantic alignment increases continuously with depth while showing only marginal improvement in final layers. Conversely, spatial discriminability initially increases but deteriorates significantly in deeper layers. Based on these findings, the authors propose three techniques to restore spatial discriminability while maintaining semantic alignment.

**Questions:**

### Questions

1. What justifies claiming minimal hyperparameter tuning when the method involves critical design choices like ATR's layer positioning and SSR's alpha parameter that directly impact performance?
2. Can you provide empirical comparisons of different abnormal token detection methods beyond Hoyer scores and alternative replacement strategies beyond Gaussian kernels ?
3. For SHE's AUC computation across datasets, are you using training or test sets, and how does this dataset dependency affect the method's practical applicability?
4. Can you provide ablation results showing SSR and SHE performance individually without ATR to better understand each component's contribution?

I am willing to increase my score if the questions above are adressed

**Ethical Concerns:**

["NO or VERY MINOR ethics concerns only"]

**Final Justification:**

I thank the authors for their detailed responses to all my raised questions. The explanations has satisfactorily addressed my initial concerns, so I will increase my score to borderline accept.

**Limitations:**

yes

**Paper Formatting Concerns:**

I don't have concerns.

**Quality:**

3

**Strengths And Weaknesses:**

## **Strengths**
**Writing Quality:** The paper is very clear and well-structured, with easily followable sections and extensive experimental validation. The ablation studies demonstrate the contribution of each proposed technique when Abnormal Token Replacement (ATR) is enabled.

**Thorough Analysis:** The comprehensive examination of CLIP's spatial discriminability and semantic alignment provides strong motivation for the proposed techniques. This analysis clearly identifies CLIP's limitations in semantic segmentation and justifies the need for the three enhancement techniques.

## **Weaknesses**
**Hyperparameters Dependency:** The paper claims minimal hyperparameter tuning requirements (L21), yet describes that ATR must be applied specifically at the penultimate layer, with earlier application degrading performance (L288-191). This layer selection appears to constitute hyperparameter tuning, contradicting the initial claim. Similarly, the alpha parameter in Spatial-Semantic Reweighting (SSR) represents another hyperparameter requiring tuning.

**Insufficient Empirical Justification:** The ATR method lacks empirical validation for key design choices, including the use of Gaussian kernels for abnormal token replacement and Hoyer scores for abnormal token detection. These seemingly arbitrary decisions require comparison against alternative detection and replacement strategies.

**Method Practicality:** Selective Head Enhancement (SHE) requires computing average AUC scores across all datasets to identify high-performing heads, making the method cumbersome and limiting its adaptability. The necessity of dataset-wide inference significantly constrains practical deployment.

**Incomplete Ablation:** Table 5's ablation study lacks individual evaluation of SSR and SHE without ATR, preventing clear understanding of each component's isolated contribution to performance improvements.

## **Clarification Points**

**Title Clarity:**  The current title lacks specificity and could be more descriptive of the actual contribution.

**Ambiguous Phrasing:**

- Lines 60-62 require clarification for better understanding.
- Line 80 needs rephrasing for improved clarity.

**Figure Readability:**  Figure 4 is difficult to interpret and would benefit from improved visualization.

**Terminology Choice:**  Line 179's use of "simple yet effective" implies simplicity as a limitation. "Simple and effective" would be more appropriate, as both qualities represent advantages.

---

> ### Author Rebuttal · Authors · 2025-07-30
>
> Thanks very much for your time and feedback on our submission. We address your concern in detail below.
>
> **W1 and Q6**. **W1**: Hyperparameters Dependency ... another hyperparameter requiring tuning. **Q6**: What justifies claiming minimal hyperparameter tuning ... directly impact performance?
>
> **A1 and A6**: Thank you for pointing this out. As the subsequent question (Q6) raises a similar concern, we address them together here. If we have misunderstood the follow-up question, we would be happy to provide additional clarification.
> While our method does introduce several hyperparameters, we would like to emphasize that these are **guided by universal patterns** we observed across datasets—such as the sparsity of abnormal tokens, the trends in SD and SA, and the presence of shared spatially discriminative heads. Thanks to these consistent phenomena, we would like to emphasize two key points:
>
> First, the hyperparameters are **generalizable** across datasets. In practice, we found that values selected on a few dataset or subset also perform well on other datasets, indicating that they are not overfitted to specific test sets. As detailed in Appendix A.2.1, we use the same set of hyperparameters for all datasets without any dataset-specific tuning.
>
> Second, the method is **robust** to a wide range of hyperparameter values, meaning that precise tuning is not required to achieve strong performance. This is demonstrated in our ablation studies (Tables 2–4), where the results remain consistent across varied hyperparameter settings.
>
> Additionally, our analysis provides **interpretable heuristics for selecting hyperparameters**. For example, in the case of an SSR starting layer, one can use the observed turning point in SD or SA as **a reliable initialization**.
>
> Taken together, while several hyperparameters are introduced, we believe these properties support the practical usability of our method without requiring extensive or fragile hyperparameter tuning.
>
> **W2 and Q7**. **W2**: Insufficient Empirical Justification ...  alternative detection and replacement strategies. **Q7**: Can you provide empirical comparisons of different abnormal token detection methods beyond Hoyer scores and alternative replacement strategies beyond Gaussian kernels?
>
> **A2 and A7**: Thank you for your feedback. As the subsequent question (Q7) raises a similar concern, we address both questions here.
> Our initial motivation for using Hoyer scores for abnormal token detection and Gaussian kernels for token replacement was based on two primary considerations:
>
> 1. Computational simplicity – both methods are lightweight, require less hyperparameter tuning, and are easy to integrate into a larger model pipeline.
> 2. Interpretability – our exploratory analysis (Figure 3 and 4) revealed that abnormal tokens tend to exhibit sparse activation patterns. Hoyer scores, as a straightforward unsupervised sparsity metric, correlated well with these tokens. Similarly, Gaussian kernel-based replacement enables smooth replacement, allowing semantic preservation while correcting local irregularities.
>
> While we acknowledge the importance of empirical comparison, our design choices were guided by these intuitive and practical considerations. We consider ATR **a flexible framework** that can accommodate other methods. To address the reviewer’s concern directly, we additionally explore the magnitude-based detection based on the observation in [1] and nearest-neighbor replacement.  The experiments are conducted based on the ClearCLIP method with ViT-B as the vision encoder.
>
> The magnitude-based detection computes the norm of each visual patch and patches above the threshold are treated as abnormal patches. Empirically, we found it can achieve best overall performance **42.5** across eight datasets when the norm threshold is set to be 14, **similar** to our sparsity-based ATR method (42.7) in Table 1. However, it is **more sensitive** than the sparsity-based method. For example, the overall performance degrade to 37.5, 41.1 and 42.0 when the norm threshold is set to 12, 13 and 15 respectively.
>
> For nearest-neighbor replacement, it replaces the abnormal token with their closest 8 neighbor patches and achieves overall performance **42.5**.  This demonstrates the flexibility of the ATR framework. In the revision, we will incorporate this discussion in the appendix.
>
> **W3**: Method Practicality: Selective Head Enhancement (SHE) ... constrains practical deployment.
>
> **A3**: We appreciate the reviewer’s comment and the opportunity to clarify. While Selective Head Enhancement (SHE) involves computing AUC scores across attention heads, it does not require full dataset-wide inference in practice.
>
> As shown in Table 4 and discussed in A1, the high-performing heads demonstrate strong generalization across datasets. Our analysis in Figure 5 further supports this: the rank distribution of AUC scores across heads reveals that certain heads consistently perform well, regardless of dataset. This consistency enables us to identify these heads using only a small subset of the datasets (e.g., a small number of samples from 3–4 out of the 8 datasets in Table 1), significantly reducing the required computational cost.
>
> Moreover, Table 4 shows that the method is robust to the number of selected heads, choosing 30 heads instead of 10 results in only a 0.1 drop in performance, suggesting that fine-grained tuning is unnecessary for practical use.
>
> Furthermore, This hyperparameter selection process reflects a one-time design choice. As detailed in Appendix A.2.1, the same set of hyperparameters once determined is used across all datasets without any modification.
>
> Taken together, these findings indicate that SHE can be implemented efficiently and remains adaptable, and we therefore believe it does not pose a significant constraint on practical deployment.
>
> **W4 and Q9**. **W4**: Incomplete Ablation: ... each component's isolated contribution to performance improvements. **Q9**: Can you provide ablation results showing SSR and SHE performance individually without ATR to better understand each component's contribution?
>
> **A4 and A9**: Thank you for your constructive question regarding ablation study presented in Table 5. As the subsequent question (Q9) raises a similar concern, we address them together here. If we have misunderstood the follow-up question, we would be happy to provide additional clarification.
>
> Due to space constraints, we distributed the ablations across multiple tables, each focusing on an individual component: ATR is analyzed in Table 2, SSR in Table 3. The ablation study of SSR without ATR is represented in Table 3. To address the reviewer’s request more directly, we additionally report the ablation results for SHE without ATR (with the number of selected heads fixed at 10, consistent with the light gray setting in Table 4): 34.0 (Context60), 37.1 (Object), 35.7 (City), 37.0  (Context59), **35.6 (Avg)**. Compared to the baseline average of **33.9**, applying only SHE yields a **1.7** point improvement, demonstrating that SHE contributes positively to performance even without the assistance of ATR.
>
> In Table 5, we focused on presenting component combinations to highlight the complementary effects of the proposed modules. For completeness, we report the performance of the combination of SSR and SHE (without ATR): 34.9 (Context60), 37.1 (Object), 36.7 (City), 37.9 (Context59), **36.6 (Avg)**. This combination outperforms applying **SSR only (34.8 avg)** and **SHE only (35.6 avg)**, reinforcing the conclusion that the proposed components are effective and complementary.
>
> In the revision, we will restructure the table and incorporate these additional ablation results to more clearly demonstrate the individual effectiveness and interplay of each component.
>
> **W5**: Clarification Points: Title Clarity... qualities represent advantages.
>
> **A5**: Thank you for your constructive feedback. We appreciate your detailed suggestions regarding the title clarity, figure readability, terminology choice, and sentence phrasing. We will revise the paper accordingly and incorporate all suggested revisions in the final version. We’re also glad to hear your thoughts on the title, and we will take them into account during the revision process.
>
> **Q8**: For SHE's AUC computation across datasets, are you using training or test sets, and how does this dataset dependency affect the method's practical applicability?
>
> **A8**: For the analysis in Figure 1, we randomly selected 1,000 samples from the training sets of each dataset to compute AUC scores across attention heads. As discussed in A2, we believe this sampling choice does not compromise the practical applicability of the method.
>
> While the exact ranking of heads may vary slightly across datasets, Figure 5 shows that high-performing heads tend to be consistent across different datasets, indicating strong generalizability. Furthermore, as shown in Table 4, we selected hyperparameters (e.g., head selection) using only a subset of datasets, and these settings generalized well to others listed in Table 1.
>
> Additionally, regarding the number of selected heads, Table 5 demonstrates that SHE is robust across a wide range of values, with minimal performance degradation when varying this parameter. Taken together, these results suggest that the dataset dependency is limited, and SHE can be applied efficiently and reliably without requiring exhaustive tuning across all datasets.
>
> [1] Darcet, T., Oquab, M., Mairal, J., & Bojanowski, P. Vision Transformers Need Registers. In The Twelfth International Conference on Learning Representations.

---

### Official Review · Reviewer_irCm · 2025-06-27

**Clarity:** 1
**Significance:** 3
**Originality:** 3
**Rating:** 3
**Confidence:** 5

**Summary:**

The paper introduces TLH-CLIP, a training-free framework that adapts CLIP for open-vocabulary semantic segmentation by leveraging spatial cues across Token, Layer, and Head levels. The authors find that: (1) some final-layer tokens degrade spatial clarity, (2) later layers favor global alignment over local detail, and (3) only a few attention heads provide strong spatial signals. Based on this, they propose three techniques—abnormal token replacement, semantic-spatial reweighting, and selective head enhancement—to improve segmentation without additional training. TLH-CLIP achieves state-of-the-art results on 8 benchmarks.

**Questions:**

- Some existing works suggest that residual connections and FFN features may negatively impact spatial localization and propose their removal. In contrast, this work proposes to upweight the residual connections. Could the authors clarify this apparent contradiction and explain why upweighting improves spatial discriminability?
- The paper uses AUC scores to select layers and heads in the Selective Head Enhancement (SHE) module. What exactly is the AUC metric used here? Since the computation relies on ground-truth labels, doesn’t this violate the training-free assumption and introduce potential data leakage? This significantly weakens the credibility of the results.
- The results in Tables 2–5 highlighted in light gray are stated to reflect the default setting, yet they differ across the tables. This inconsistency may confuse readers. Could the authors clarify whether these are truly default settings and explain the discrepancy?
- The proposed method involves several hyperparameters. It appears that performance gains may depend heavily on careful parameter tuning. How are these parameters selected under a fully unsupervised and training-free setting?
- The authors state that ATR is applied only to the penultimate layer. However, Equation (9) appears to apply ATR to features from multiple selected layers.

**Ethical Concerns:**

["NO or VERY MINOR ethics concerns only"]

**Final Justification:**

I am inclined to reject this paper due to its unclear figure illustrations, questionable Selective Head Enhancement (SHE) module, and inconsistent parameter sensitivity results.

**Limitations:**

yes

**Quality:**

2

**Strengths And Weaknesses:**

Strengths
- **Originality**: The proposed techniques are conceptually simple yet introduce novel insights into exploiting CLIP's internal structures (tokens, layers, heads) for dense prediction tasks.
- **Significance**:  TLH-CLIP demonstrates substantial performance gains over selected baselines across multiple benchmarks.

Weaknesses
- **Clarity**: The quality of the visualizations is very poor. For instance, Figure 4 is barely legible due to excessively small text, even when zoomed in. This significantly hinders the reader's understanding.
- **Quality**: Certain technical aspects lack clarity and rigor. For example, the use of ground-truth labels to compute the AUC metric in Selective Head Enhancement (SHE) raises concerns, especially in a training-free setting where no supervision should be assumed.

---

> ### Author Rebuttal · Authors · 2025-07-30
>
> Thanks very much for your time and feedback on our submission. We address your concern in detail below.
>
> **W1**: Clarity: The quality of the visualizations ... understanding.
>
> **A1**: Thank you for pointing this out. We apologize for the poor readability of Figure 4. In the revised version, we will improve the figure by enlarging the text, increasing resolution, and optimizing layout to ensure better readability.
>
> **W2, second question in Q4** and **Q6**. **W2**: Quality: Certain technical aspects lack clarity and rigor... a training-free setting where no supervision should be assumed. **second question in Q4**: Since the computation ... the results. **Q6**: The proposed method involves several hyperparameters ... a fully unsupervised and training-free setting?
>
> **A2, second part of A4 and A6**: We appreciate the reviewer’s feedback and the opportunity to clarify this point. We believe the reviewer raises a similar concern in the second question of Q4 and in Q6; therefore, we address them together here. If we have misunderstood the reviewer’s follow-up question, we would be happy to provide further clarification.
>
> The term **“training-free”** in our paper refers to the fact that we do **not perform any gradient-based optimization or parameter updates** on the pretrained CLIP model. From my understanding, this definition is **orthogonal** to whether or not supervision is involved or the **concept of unsupervised**. Our method only modifies the inference procedure of the vision encoder to enhance its outputs, and does **not require any additional training**.
>
> As for the use of supervision, **we do not claim that our method is entirely free of supervision**. Rather, we clarify that ground-truth labels are **used only for evaluation** and for **hyperparameter selection on a small subset of the dataset**. (The reviewer’s concern about“ fully unsupervised” in Q5). This practice is **consistent with many prior works** in open-vocabulary semantic segmentation. For instance, NACLIP [1] also selects hyperparameters (e.g., Gaussian window parameters) based on performance on a small set of evaluation datasets.
>
> As for the reviewer’s concerns regarding potential data leakage (Q4) and hyperparameter sensitivity (Q6), we would like to clarify that, as demonstrated in our ablation study, the selected hyperparameters are **robust** across a wide range of values, indicating that precise tuning is not critical for achieving strong performance. Furthermore, these hyperparameters are highly **generalizable**: they are selected using a small, minimally supervised evaluation set and then fixed across a broad range of tasks and datasets, consistently yielding significant performance improvements. As detailed in Appendix A.2.1, **the same set of hyperparameters is used across all datasets without any modification**. This hyperparameter selection process reflects a one-time design choice rather than per-dataset optimization.
>
> Importantly, this minimal use of supervision, combined with the fact that our method requires **no additional training**, helps preserve the **generalizability** of the CLIP model and results in **less risk of data leakage** compared to fine-tuning-based methods. The CLIP encoder remains frozen throughout, and our approach involves **only 5–6 hyperparameters**, which are determined using a small validation subset. In contrast, as discussed in Lines 31–35, methods that rely on **fine-tuning may risk overfitting to the limited categories or supervision** used during training, potentially undermining CLIP’s pretrained generalization capabilities. Our approach is designed to retain this generality while improving spatial discriminability and semantic alignment in a lightweight, plug-and-play manner.
>
> We hope this clarification addresses the reviewer’s concerns and helps illustrate the principled design and practical benefits of our training-free, minimally supervised method.
>
> **Q3**:Some existing works ... explain why upweighting improves spatial discriminability?
>
> **A3**: We appreciate the reviewer’s question and the opportunity to clarify this point. We believe there is **no contradiction between our approach and prior works**.
>
> While some existing methods, such as ClearCLIP, NaCLIP, and related works, suggest that residual connections and FFN features may harm spatial localization, these approaches only remove such components **at the final transformer layer**(e.g., layer 12 for ViT-B or layer 24 for ViT-L). In fact, these works do not apply such modifications to earlier layers.
>
> Our empirical findings support this design choice: removing residual connections and FFN modules from earlier layers (e.g., layers 1–11 for ViT-B or 1–23 for ViT-L) leads to significant performance degradation. For example, even removing the residual connection and FFN from only the penultimate layer (layer 11 in ViT-B or layer 23 in ViT-L) results in a substantial drop in performance. This is because removing these components at earlier stages disrupts the learned feature representations and causes distributional shifts in the input to subsequent layers.  To illustrate, let $x^{11}$ and $\tilde{x}^{11}$ represent the outputs of layer 11 of ViT-B with and without residuals, respectively. The difference between these two representations can be large, leading to mismatched inputs for layer 12, which in turn weakens the alignment between visual tokens and the text embeddings. This explains why prior works limit such removals to the final layer only.
>
> In contrast, our SSR method takes a complementary approach: we explore whether **modifying the layers preceding the final layer** can improve performance, motivated by the observations in Figure 1. This direction is orthogonal to prior work, which primarily focuses on the final layer. To the best of our knowledge, our work is the first to systematically investigate modifications to the inference procedure of earlier layers as a means to enhance spatial discriminability and overall performance.
>
> The rationale for upweighting the residual connection is based on the observation in Figure 1: spatial discriminability follows an inverted U-shaped curve across layers, while semantic alignment increases monotonically but saturates near the final layers. This suggests that the visual tokens features middle-to-late layers become more entangled and less discriminative near the end.Therefore, we conjecture that improving the spatial discriminability is one of the main constraints that limits the overall performance.  According to Equations (1) and (2) in our paper, the intermediate feature representation consists of a residual path (carrying information from earlier layers) and outputs from attention and MLP modules. By slightly upweighting the residual component (e.g. $\alpha=0.1$), we aim to enhance spatial discriminability by amplifying information from earlier, more spatially discriminative layers, without significantly disturbing the overall distribution to prevent degrading semantic alignment.
>
> **First question in Q4**: The paper uses AUC scores ... What exactly is the AUC metric used here?
>
> **First part of A4**: We apologize for the omission of these details and will clarify them in the revision.  The AUC metric stands for “Area Under the Curve”, referring specifically to the area under the ROC (Receiver Operating Characteristic) curve. It is a widely used metric for binary classification tasks that measures how well a model can distinguish between two classes. A higher AUC value indicates stronger discriminative capability.
>
> In our setting, we follow a common approach used in prior work: for any pair of visual token features, we compute the cosine similarity between them and treat this similarity as the input feature for a binary classification task. The label is set to 1 if both patches come from the same category, and 0 otherwise. This framing allows us to evaluate how well the representations are discriminative.
>
> **Q5**:The results in Tables 2–5 highlighted in light gray ... default settings and explain the discrepancy?
>
> **A5**: The light gray highlights are not meant to represent a single default configuration; rather, **they denote the best-performing settings under different hyperparameter choices for each experiment**.
>
> As clarified in lines 281–284, for Tables 2, 3, and 5, the baseline refers to using SCLIP with the standard $QK^\top$ attention replaced by the combination $QQ^\top + KK^\top$, without incorporating our proposed improvements. For Table 4, as discussed in lines 305–308, we empirically found that SHE performs best when combined with ATR; hence, we use the best-performing ATR configuration as the baseline in this case. We hope this clarification resolves your confusion.
>
> **Q7**: The authors state that ATR is applied only to the penultimate layer... multiple selected layers.
>
> **A7**: We thank the reviewer for pointing out this confusion. When we state that “ATR is applied only to the penultimate layer,” we are referring to the fact that only the output features from the penultimate layer are modified, which in turn affects the inputs to the subsequent final layer.
>
> Eq. (9) is used to compute the spatial discriminability of each head in earlier layers. For that goal, we remove the abnormal tokens to better capture the spatial discriminability. But once these are computed, we do **not alter the forward computation of any following layers prior to the final layers**. Instead, the information of spatial discriminability of each head is **only used to refine the final-layer features** according to the expression at the end of section 3.
>
> To prevent further confusion, we will rename and clarify the notations in Eq. (9) and the surrounding text in the revision to more clearly distinguish between feature extraction versus feature modification during inference.
>
> [1] Hajimiri et al.,  "Pay attention to your neighbours: Training-free open-vocabulary semantic segmentation." WACV.  2025.

---

> > ### Author Response · Authors · 2025-08-05
> > **Inquiry for further clarification**
> >
> > Dear Reviewer irCm, thank you again for your time and consideration of our rebuttal. We hope our responses have addressed your concerns satisfactorily. If there are any remaining issues or points that need further clarification, please don’t hesitate to let us know and we would be happy to provide additional explanations. We truly appreciate your feedback and support.

---

> ### Comment · Reviewer_irCm · 2025-08-05
>
> Thank you to the authors for their thorough and detailed response.
>
> - AUC scores
>
> `The label is set to 1 if both patches come from the same category, and 0 otherwise.`
>
> This implies that the AUC scores are computed based on ground-truth semantic labels.
>
> In lines 232–233, the authors mention:
>
> `Let AUC$^s_{l,h}$ denote the AUC score of the representations from the h-th head in the l-th layer for datase $s\in${VOC, Context, ADE, Stuff}`
>
> This suggests that ground truth from multiple datasets is used to compute the AUC scores.
>
> Could the authors clarify whether it is reasonable to use ground-truth labels from various datasets?
>
> - Ablation study
>
> `the same set of hyperparameters is used across all datasets without any modification. `
>
> Given this, it would be valuable to see an analysis of how sensitive the method is to these hyperparameters. Why not conduct ablation experiments around this default setting (In Tables 2, 3, and 4) to better understand the impact of each hyperparameter on performance?

---

> ### Author Response · Authors · 2025-08-05
> **Response to reviewer  irCm**
>
> We thank you again for your time and consideration of our rebuttal.
>
> Regarding your follow-up question about whether it is reasonable to use ground-truth labels from various datasets, we are not entirely sure we fully understand your concern. Our current guess is that you may be asking whether using ground-truth labels from different datasets could potentially lead to data leakage. If this is not your intended point, we would greatly appreciate further clarification so we can address it more precisely.
>
> If this is your intended point, we would like to offer the following clarification:
>
> First, the AUC score is indeed based on ground-truth labels. The use of AUC to analyze patch discriminability was first proposed in [1] and has since been widely adopted in subsequent work for such analysis.
>
> In our case, we compute AUC across multiple datasets **primarily for analysis and illustration** purposes—specifically to demonstrate that certain attention heads exhibit **consistent and generalizable** spatial discriminability across datasets. While the exact ranking of heads may vary slightly between datasets, several heads (e.g., (8, 9), (9, 12), and (9, 5)) consistently appear among the top 15 most discriminative heads, as shown in Figure 5.
>
> Second, as shown in the ablation study in Table 4, the number of selected heads is **robust across a wide range**. For example, increasing the number of heads from 10 to 30 results in only a 0.1-point drop in average performance. This suggests that carefully determining the exact number of selected heads is unnecessary.
>
> Take these two point together, it indicates that selecting heads based on **a few number of subsets from multiple datasets is sufficient** to enhance the spatial discriminability of features in the later transformer blocks, and this **generalizes well across the remaining datasets**.
>
> Finally, it is worth noting that for Figure 5, we use 1,000 samples from the **training set of each dataset** for illustration and hyperparameter selection purposes. Once the hyperparameters are selected using these training sets, the model demonstrates strong generalization performance across various **test** datasets—including those whose training sets not used for hyperparameter selection—as shown in Table 1.
>
> [1] Mukhoti, Jishnu, et al. "Open vocabulary semantic segmentation with patch aligned contrastive learning." Proceedings of the IEEE/CVF Conference on Computer Vision and Pattern Recognition. 2023.

---

> ### Comment · Reviewer_irCm · 2025-08-05
>
> `we compute AUC across multiple datasets primarily for **analysis** and **illustration** purposes.`
>
> and
>
> `it is worth noting that for Figure 5, we use 1,000 samples from the training set of each dataset for illustration and **hyperparameter selection** purposes.`
>
> This seems contradictory—were the AUC scores used solely for analysis and illustration, or were they also used for hyperparameter tuning?

---

> ### Author Response · Authors · 2025-08-05
> **Response to Reviewer irCm**
>
> They are also used for hyperparameter tuning. We would like to clarify that the term “primarily” does not imply “exclusively” or “solely”. Moreover, in this context, it emphasizes the AUC computation, rather than multiple datasets.
>
> For details regarding the ablation study setup, please refer to lines 280–284 and 305–308. The setup for Table 1 is described in lines 255–266, with additional details provided in Appendix A.2.1.
>
> In response to your updated question:
>
> “Given this, it would be valuable to see an analysis of how sensitive the method is to these hyperparameters. Why not conduct ablation experiments around this default setting (in Tables 2, 3, and 4) to better understand the impact of each hyperparameter on performance?”
>
> The similar question has been raised by Reviewer DEqc (comments A4 and A9) and Reviewer tN1T (comment A3). For your convenience, we include our response to them below:
>
> Due to space constraints, we distributed the ablations across multiple tables, each focusing on an individual component: ATR is analyzed in Table 2, SSR in Table 3. The ablation study of SSR without ATR is represented in Table 3. To address the reviewer’s request more directly, we additionally report the ablation results for SHE without ATR (with the number of selected heads fixed at 10, consistent with the light gray setting in Table 4): 34.0 (Context60), 37.1 (Object), 35.7 (City), 37.0 (Context59), 35.6 (Avg). Compared to the baseline average of 33.9, applying only SHE yields a 1.7 point improvement, demonstrating that SHE contributes positively to performance even without the assistance of ATR.
>
> In Table 5, we focused on presenting component combinations to highlight the complementary effects of the proposed modules. For completeness, we report the performance of the combination of SSR and SHE (without ATR): 34.9 (Context60), 37.1 (Object), 36.7 (City), 37.9 (Context59), 36.6 (Avg). This combination outperforms applying SSR only (34.8 avg) and SHE only (35.6 avg), reinforcing the conclusion that the proposed components are effective and complementary.
>
> In the revision, we will restructure the table and incorporate these additional ablation results to more clearly demonstrate the individual effectiveness and interplay of each component.

---

> > ### Author Response · Authors · 2025-08-07
> > **Official Comment by Authors**
> >
> > Thank you very much for your questions. We hope the above explanations address your concerns and look forward to further discussions and feedback. If you need more clarification, please let us know.

---

### Official Review · Reviewer_tN1T · 2025-07-01

**Clarity:** 2
**Significance:** 3
**Originality:** 3
**Rating:** 5
**Confidence:** 3

**Summary:**

This paper focuses on improving the ability of CLIP models to perform semantic segmentation without extra training. The authors introduce a new framework called TLH-CLIP that enhances CLIP’s spatial discriminability across three levels: tokens, layers, and attention heads. They use three techniques: abnormal token replacement (ATR), spatial-semantic reweighting (SSR), and selective head enhancement (SHE). ATR identifies and smooths out high-activation tokens that disrupt spatial coherence. SSR downweighs attention/MLP path and upweights residual paths in the final few (before last) layers. SHE uses specific attention heads that are good at capturing spatial details to improve the output. The results show that TLH-CLIP outperforms other training-free methods on common semantic segmentation benchmarks, achieving top performance when paired with ResCLIP.

**Questions:**

1.	Issues mentioned in the weaknesses section.
2.	How does TLH perform on vanilla CLIP (or MaskCLIP if you need such changes to the last layer)?
3.	Fig. 1 is interesting. Considering part (e), do you suggest that for that backbone and dataset, we would get better performance if we simply removed layers 20-23, and fed the output of layer 19 directly to layer 24? Have you tried doing something like this to see whether the SD and SA metrics are indeed aligned with actual performance?
4.	In Fig. 2, is the red dot the anchor patch from which you are calculating the attention? If so, this should be mentioned in the caption.

I'm willing to increase my score should the authors address my concerns. The main ones are: discussions in Section 2.3, ablation studies of the 3 components, adding citations, performance on vanilla CLIP, actual performance correlation of SD and SA. About the hyperparameters, I don't think the authors can do much.

**Ethical Concerns:**

["NO or VERY MINOR ethics concerns only"]

**Final Justification:**

I think the paper can be useful to the OVSS community and hence I am more leaning toward accepting it.

**Limitations:**

There shouldn't be potential negative societal impacts for this work. As for the limitations, I do not recall explicit discussion about it in the paper.

**Quality:**

3

**Strengths And Weaknesses:**

### Strengths

1.	The paper offers a fresh look at how CLIP works internally and finds that final layers prioritize semantic alignment over spatial detail, among other observations.
2.	The proposed approach is modular and can be applied on top of existing methods.
3.	Their model significantly improves the performance and achieves SOTA on multiple benchmarks.
4.	The model shows robustness in performing well on the base and large backbones.


### Weaknesses

1.	Too many hyperparameters are introduced in this paper: tau, sigma, alpha, l_start, l_end, k, beta. Some of the values might have been tuned on test datasets.
2.	I’m not fully convinced by the evidence with which Section 2.3’s conclusion was made, after slightly reading about the hoyer score for the first time. I’ll follow other reviewers’ insight on this. If they find the arguments convincing, the conclusion is rather important and transformational as I believed the claims in [33] to be true.
3.	Ablation studies in Table 5 are useful, but makes me wonder why not all the combinations have been included? Also, please align the mIoU in the table to the corresponding text (37.4 vs 37.5).
4.	Spatial discriminability and semantic alignment are mentioned repeatedly in the intro, without indications of what they actually mean up until Section 2.1. Also, the definition of spatial discriminability in L106 doesn’t seem to incorporate the “spatial” characteristics of tokens (their relative location), hence the naming is a bit confusing (but no need to change it).
5.	Citation:
	1.	When talking about high-magnitude tokens, e.g., in the intro, proper citation to [33] is needed.
	2.	L283: I believe removing the residual connections and FFN was discussed in [16] before [15] and [20].

---

> ### Author Rebuttal · Authors · 2025-07-30
>
> We appreciate your thoughtful comments and the opportunity to clarify and improve our work. We address your concern in detail below.
>
> **W1**: Too many hyperparameters ... tuned on test datasets.
>
> **A1**: While our method does introduce several hyperparameters, we would like to emphasize that these are guided by **universal patterns** we observed across datasets—such as the sparsity of abnormal tokens, the trends in SD and SA, and the presence of shared spatially discriminative heads. Thanks to these consistent phenomena, we would like to emphasize two key points:
>
> First, the hyperparameters are **generalizable** across datasets. In practice, we found that values selected on a few dataset or subset also perform well on other datasets, indicating that they are not overfitted to specific test sets. As detailed in Appendix A.2.1, we use the same set of hyperparameters for all datasets **without any dataset-specific tuning**.
>
> Second, the method is **robust** to a wide range of hyperparameter values, meaning that precise tuning is not required to achieve strong performance. This is demonstrated in our ablation studies (Tables 2–4), where the results remain consistent across varied hyperparameter settings.
>
> Additionally, our analysis provides **interpretable heuristics** for selecting hyperparameters. For example, in the case of an SSR starting layer, one can use the observed turning point in SD or SA as a **reliable initialization**.
>
> Taken together, while several hyperparameters are introduced, we believe these properties support the practical usability of our method without requiring extensive or fragile hyperparameter tuning.
>
> **W2**: I’m not fully convinced ... claims in [33] to be true.
>
> **A2**: Thank you for your feedback. We are not entirely certain whether your concern pertains to the validity of the evidence presented or the interpretation of our conclusions, and we would be happy to provide further clarification if needed.
>
> Regarding our conclusion, we would like to emphasize that our findings do not contradict prior results suggesting that abnormal tokens primarily capture linear-separable features. Rather, we aim to enrich and extend this understanding. To illustrate, consider a simplified example with abnormal tokens from four distinct classes—cat at [101, 100], dog at [99, 100], train at [100, 101], and airplane at [100, 99]. The shared global mean of [100, 100] dominates each representation, resulting in high similarity across tokens, despite differing class labels. This suggests that abnormal tokens predominantly encode class-agnostic, shared features, with a global bias that can obscure class-specific variations. In particular, when attention is computed with other tokens, these abnormal tokens become largely indistinguishable.
>
> For the experiment results, we became aware of a contemporary paper [1] that reports a similar phenomenon regarding abnormal tokens, after our submission. However, this work was made publicly available only after the NeurIPS submission deadline.
>
> Specifically, in Section 3.1 of their paper, they also observe that abnormal tokens exhibit sparse and large activations, which aligns with our Hoyer score analysis used to quantify activation sparsity. Moreover, in Section 3.2 ("Detecting register neurons"), “neurons whose average activation at outlier positions is consistently high across images”, which also indicates certain neurons consistently appear high activated across images.
> While their observations overlap with some aspects of our findings, our work offers a complementary and more comprehensive analysis. In particular, we explicitly compare abnormal patches across patch, layer, and image levels, and show that these abnormal patches exhibit extremely high cosine similarity (greater than 0.998) across the whole ImageNet validation set.
>
> We believe this opens an interesting direction for future research into the role of global biases in vision-language models. For example, considering the richness of image content, which may not be fully captured by textual description during CLIP-style training, we hypothesize that the global bias may serve as a proxy for broad visual semantics, aligning with prompts like “a photo of [blank]”. This intriguing possibility lies beyond the scope of our current study and is left for future exploration. To support further research in this area and support our discovery, we are happy to share our code for computing similarity with the community.
>
> **W3**: Ablation studies in Table 5 are useful ...  (37.4 vs 37.5).
>
> **A3**: Thank you for your interest in the ablation study presented in Table 5. Due to space constraints, we distributed the ablations across multiple tables, each focusing on an individual component: ATR is analyzed in Table 2, SSR in Table 3. For SHE, we observed that it becomes beneficial primarily when abnormal tokens are replaced with ATR, as shown in Table 4 and discussed in the section “Analysis of the number of selected heads.”
>
> To address the reviewer’s request regarding the individual role of SHE more directly, we additionally report the ablation results for **SHE without ATR** (with the number of selected heads fixed at 10, consistent with the light gray setting in Table 4): 34.0 (Context60), 37.1 (Object), 35.7 (City), 37.0  (Context59), 35.6 (Avg). Compared to the baseline average of **33.9**, applying only SHE yields a **1.7 point improvement**, demonstrating that SHE contributes positively to performance even without the assistance of ATR.
>
> In Table 5, we focused on presenting component combinations to highlight the complementary effects of the proposed modules. For completeness, we report the performance of the combination of SSR and SHE (without ATR): 34.9 (Context60), 37.1 (Object), 36.7 (City), 37.9 (Context59), **36.6** (Avg). This combination outperforms applying SSR only (**34.8 avg**) and SHE only (**35.6 avg**), reinforcing the conclusion that the proposed components are complementary.
> In the revision, we will restructure the table and incorporate the additional ablation results to more clearly demonstrate the individual effectiveness and interplay of each component, and fix the typo in the updated version.
>
> **W4**: Spatial discriminability ... a bit confusing (but no need to change it).
>
> **A4**: We appreciate your constructive comments and suggestions. We will revise the introduction to briefly explain their intuitive meanings upfront and guide readers toward the formal definitions in Section 2.1.
>
> Regarding the naming of spatial discriminability, we understand your concern after your explanation. In the current submission, this term is intended to describe the discriminability between different spatial visual tokens in ViT. We will make this interpretation clearer or rename it in the revised version to avoid confusion.
>
> **W5**: Citation... before [15] and [20].
>
> **A5**: We appreciate the reviewer’s detailed citation suggestion. We will revise the manuscript to include the appropriate citation to [33] in the introduction and update the references in Line 283.
>
> **Q6**: How does TLH perform on vanilla CLIP?
>
> **A6**：During the rebuttal period, we conducted additional experiments on the vanilla CLIP model with the ViT-B architecture to evaluate the generalizability of our method. The results demonstrate that our proposed TLH method remains effective even without modifications to the final layer. Specifically, TLH improves the overall performance from **11.6** (as reported in Table 1 of our paper) to **17.5**. The detailed results across datasets are as follows: 23.0(VOC21),  11.6(Context60),  14.3(Object),  50.3(VOC20),  11.4(City), 15.9(Context59),  4.8(ADE), 8.7(Stuff), 17.5(Avg).
>
> **Q7**：Fig. 1 is interesting... aligned with actual performance?
>
> **A7**: Thank you for your interest in figure 1. While simply removing layers 20-23, and feeding the output of layer 19 directly to layer 24 leads to better performance than baseline, we empirically found that our SSR method, which upweights the residual components at final layers in equation (9), consistently outperforms this approach.
>
> For instance, using ClearCLIP as the base method with ATR and SHE incorporated, simply skipping layers 20–23 in ViT-L vision encoder yields an average performance of 40.8, which is better than the baseline ClearCLIP (34.9), but still lower than our full TLH-CLIP method (42.7), as shown in Table 6. Detailed results across datasets are: 60.3(VOC21), 33.7(Context60), 32.1(Object), 77.3(VOC20), 40.8(City), 38.8(Context59), 20.1(ADE), 23.7 (Stuff). We will add this result in the revision either in the main paper or the appendix.
>
> This is also supported by Figure 7 in the appendix, which shows that applying SSR to intermediate layers enhances SD and SA in later layers, surpassing what is achievable by relying solely on earlier-layer outputs. As a result, SSR leads to better segmentation performance.
>
> For your second question: we empirically observed that high values of both SD and SA are strong indicators of good overall performance. However, a high value in only one of them does not necessarily guarantee strong performance. To clarify this relationship, we will include layer-wise segmentation performance in the revised version of the paper, plotted alongside SD and SA metrics. Since both factors jointly influence final performance, we believe this combined visualization will help convey the interaction more clearly. If the reviewer has suggestions for a more effective way to present these results, we would be happy to incorporate them.
>
> **Q8**: In Fig. 2, is the red dot ... in the caption.
>
> **A8**: Yes, the red dot in Figure 2 represents the anchor patch from which the attention is calculated. We will follow your suggestion to revise the figure caption in the updated version.
>
> [1] Jiang, Nick, et al. "Vision Transformers Don't Need Trained Registers." arXiv preprint arXiv:2506.08010 (2025).

---

> > ### Comment · Reviewer_tN1T · 2025-08-02
> >
> > I thank the authors for the detailed rebuttal. Many of my concerns have been addressed, and as I am leaning toward accepting the paper, I increase my score.

---

> > > ### Author Response · Authors · 2025-08-02
> > > **Response to Reviewer tN1T**
> > >
> > > Thank you very much for your thoughtful and constructive comments raised before, which have been helpful in improving the quality of our paper. We appreciate your consideration of our rebuttal and are glad to hear that many of your concerns have been addressed. Thank you also for your willingness to increase your score.

---

### Official Review · Reviewer_WdtL · 2025-07-02

**Clarity:** 2
**Significance:** 2
**Originality:** 2
**Rating:** 4
**Confidence:** 3

**Summary:**

The paper does an analysis on the spatial discriminability issue of CLIP models for open-vocabulary segmentation. The authors introduce three training-free techniques which reduce this issue: abnormal token replacement, spatial-semantic reweighting and selective head enhancement. The first works at the token level, the second on the layer level and the third on the head level. They not only adapt the final layer, but also earlier layers. The techniques are applicable to several existing open-vocabulary segmentation methods and improve performance by 3.5 miou on average.

**Questions:**

- A CLIP model wasn't trained to make its patch token embeddings aligned with language. It's not clear why using patch token embeddings at all would work, as I would expect only the CLS token is aligned with language and produces logical cosine similarities with text encoded class embeddings?

**Ethical Concerns:**

["NO or VERY MINOR ethics concerns only"]

**Final Justification:**

I will keep my score at borderline accept. The method lacks scalability, addressing symptoms rather than the core problems, and appears to involve questionable use of data for hyperparameter tuning. However, the results may still offer useful insights for future work that tackles the stated issues more fundamentally.

**Limitations:**

yes

**Paper Formatting Concerns:**

line 80, 142 have some issues

**Quality:**

3

**Strengths And Weaknesses:**

Strengths:
- The improved performance resulting from the techniques applied in this paper provide valuable insights in the shortcomings of CLIP models in their spatial discriminability and alignment.
- Splitting the problem up in spatial discriminability and alignment makes it easier to study the problem.
- The problem analysis is elaborate, at the token, head and layer levels.
- The insight that global tokens might act as biases is interesting and might help in explaining the global token phenomenon.

Weaknesses:
- The techniques applied are like model surgery, although they have empirical benefits, they are human designed and thus not very scalable in the long run (e.g. selecting specific layers, specific tokens). They don't fully solve the problem. In the end, a more fundamental solution to the problem of poor spatial discriminability and alignment is needed, and this paper does not provide it yet, though might bring us closer through its insights on the existing problems.
- The Abnormal token replacement module is error-prone. If neighbouring patches also classify wrongly, so will the reconstructed rejected patch. Also, if the neighbours truly should correspond to a different class, this will also wrongly classify the reconstructed rejected patch. There is no flexibility in this approach to solve these edge cases.
- The spatial-semantic reweighting makes later blocks contributions to the residual stream smaller. It is however never verified if a small amount of contribution is better than no contribution at all, by simply selecting an earlier layer output to do segmentation (similar to the Perception Encoder paper by Bolya et al in 2025).
- The selective head enhancement section is hard to understand.
- There are several hyperparameters which require careful tuning, which makes the method less plug-and-play.
- Besides ViT-L, where the method does not improve upon ViT-B, no other CLIP models are tried. It is unclear how the method would perform on other CLIP models, such as CLIP with registers or Perception Encoder, which both aim to solve the global token issue in different ways.
- Improvements for MLLMs are mentioned but never experimentally verified.

---

> ### Author Rebuttal · Authors · 2025-07-30
>
> Thank you very much for your time and valuable feedback on our submission. We appreciate your thoughtful comments and the opportunity to clarify and improve our work. We address your concern in detail below.
>
> **W1**: The techniques applied are like model surgery, although they have empirical benefits, they are human designed and thus not very scalable in the long run (e.g. selecting specific layers, specific tokens). They don't fully solve the problem. In the end, a more fundamental solution to the problem of poor spatial discriminability and alignment is needed, and this paper does not provide it yet, though might bring us closer through its insights on the existing problems.
>
> **A1**: We thank the reviewer for this insightful comment. The main goal of our work is to provide a detailed understanding of the causes of poor spatial discriminability and semantic alignment in current CLIP models, which are widely used as vision backbones. Through our analysis, we identify the emergence and behavior of abnormal tokens, showing that they are sparse yet dominant and class-agnostic, leading to representational collapse and misalignment. This analysis is very computationally efficient, and building on these findings, our proposed approach is also computationally efficient (e.g., selecting specific layers and tokens incurs negligible overhead) and can be easily applied to a wide range of models, making it scalable in practice.
>
> That said, we agree with the reviewer that the root causes of poor spatial discriminability and semantic alignment ultimately stem from the pretraining paradigm of CLIP-like vision–language models. Addressing these issues fundamentally may require rethinking training objectives, data composition, or model architectures. We believe our analysis provides valuable insights that can inform the design of next-generation models with stronger alignment and spatial reasoning capabilities. We will add this discussion in the revision.
>
> **W2**: The Abnormal token replacement module is error-prone. If neighbouring patches also classify wrongly, so will the reconstructed rejected patch. Also, if the neighbours truly should correspond to a different class, this will also wrongly classify the reconstructed rejected patch. There is no flexibility in this approach to solve these edge cases.
>
> **A2**: We agree with the reviewer that the abnormal token replacement module may be error-prone, particularly when neighboring patches are themselves misclassified or belong to a different class. This concern was a key motivation behind our decision to adopt the Gaussian smoothing approach rather than a simple nearest-neighbor replacement. This method strikes a balance between mitigating the risk of overfitting to noisy neighbors and preserving the local semantic structure of the image. We will incorporate this discussion in the revision. We are very interested in exploring more adaptive or confidence-aware strategies and would welcome any suggestions from the reviewer for future iterations of this work.
>
> **W3**: The spatial-semantic reweighting makes later blocks contributions to the residual stream smaller. It is however never verified if a small amount of contribution is better than no contribution at all, by simply selecting an earlier layer output to do segmentation (similar to the Perception Encoder paper by Bolya et al in 2025).
>
> **A3**: Thank you for the constructive suggestion. While directly using earlier layer outputs can offer some improvement over baseline CLIP performance, we empirically found that our SSR method consistently outperforms this approach.
> For instance, using ClearCLIP as the base method with ATR and SHE incorporated, simply skipping layers 20–23 in ViT-L vision encoder and and feeding the output of layer 19 directly to layer 24,  yields an average performance of 40.8, which is better than the baseline ClearCLIP (34.9), but still lower than our full TLH-CLIP method (42.7), as shown in Table 6. Detailed results across datasets are: 60.3(VOC21), 33.7(Context60), 32.1(Object), 77.3(VOC20), 40.8(City), 38.8(Context59), 20.1(ADE), 23.7 (Stuff). We will add this result in the revision either in the main paper or the appendix.
>
> This is also supported by Figure 7 in the appendix, which shows that applying SSR to intermediate layers enhances spatial discriminability (SD) and semantic alignment (SA) in later layers, surpassing what is achievable by relying solely on earlier-layer outputs. As a result, SSR leads to better segmentation performance.
>
> **W4**:The selective head enhancement section is hard to understand.
>
> **A4**: We thank the reviewer for pointing out this confusion, and we will revise this section to improve its clarity and presentation
> Intuitive speaking, the SHE is motivated by recent findings [1], which show that different attention heads capture different visual concepts such as color, shape, and texture. This inspired us to investigate whether certain heads are more effective at capturing discriminative spatial information relevant for downstream tasks like segmentation. As shown in Figure 5, some attention heads consistently exhibit high discriminability across different datasets. Based on this observation, rather than treating all heads equally, we selectively extract features from the most discriminative heads and use them to enhance the output features produced by CLIP's vision encoder.
>
> **Q5**: There are several hyperparameters which require careful tuning, which makes the method less plug-and-play.
>
> **A5**: We appreciate the reviewer’s comment. While our method does introduce several hyperparameters, we would like to emphasize that these are guided by generalizable and consistent patterns we observed across datasets—such as the sparsity of abnormal tokens, the trends in SD and SA, and the presence of shared spatially discriminative heads.
>
> Thanks to these consistent phenomena, the selected hyperparameters generalize well across datasets. As detailed in Appendix A.2.1, we use the same set of hyperparameters for all datasets without any dataset-specific tuning. Moreover, our ablation studies (Tables 2–4) show that the method is robust to a wide range of hyperparameter values, indicating that precise tuning is not necessary to achieve strong performance.
>
> Additionally, our analysis provides interpretable heuristics for selecting hyperparameters. For example, in the case of an SSR starting layer, one can use the observed turning point in SD or SA as a reliable initialization.
>
> Taken together, we believe these properties support the practical usability of our method without requiring extensive or fragile hyperparameter tuning.
>
> **Q6-Q7**: Besides ViT-L, where the method does not improve upon ViT-B, no other CLIP models are tried. It is unclear how the method would perform on other CLIP models, such as CLIP with registers or Perception Encoder, which both aim to solve the global token issue in different ways. Improvements for MLLMs are mentioned but never experimentally verified.
>
> **A6-A7**: We appreciate the reviewer’s thoughtful comment. We agree that evaluating the generalizability of our method beyond CLIP and validation of the benefits on MLLM would enhance the broader impact of our work. However, our current implementation is based on the NaCLIP codebase using the MMPretrain framework, which does not yet support the suggestions. Due to the time constraints of the rebuttal period, we were unable to complete the necessary integration. We will implement the required components and provide the evaluation results during the discussion phase.
>
> Thank you for your understanding and thoughtful feedback.
>
> **Q8**: A CLIP model wasn't trained to make its patch token embeddings aligned with language. It's not clear why using patch token embeddings at all would work, as I would expect only the CLS token is aligned with language and produces logical cosine similarities with text encoded class embeddings?
>
> **A8**: We thank the reviewer for raising this important point. While it is true that CLIP is trained to align the CLS token with the text embedding, prior work and our own empirical analysis suggest that patch-level embeddings also capture semantically meaningful information, even though they are not explicitly supervised to align with text during training.
>
> This phenomenon arises because, during CLIP's contrastive pretraining, the vision encoder must learn to extract discriminative visual features that contribute to the global image representation. These features are naturally distributed across spatial patches. As a result, although patch tokens are not directly aligned with language embeddings, they frequently reflect spatial regions that correspond to semantically relevant visual concepts associated with the image-level text.
>
> Several studies [1, 2] have shown that patch activations in CLIP correlate with object locations and visual concepts. Our findings in Figure 1 further support this: patch tokens exhibit increasing semantic alignment with text embeddings across layers, reaching saturation in the final layers.
>
> Therefore, even though patch tokens are not individually aligned with language during training, they still encode class-relevant, semantically rich information. This property enables CLIP to serve as a strong foundation for open-vocabulary segmentation and other dense prediction tasks. We will incorporate some the above discussions in the revision.
>
> [1] Gandelsman, Yossi, Alexei A. Efros, and Jacob Steinhardt. "Interpreting CLIP's Image Representation via Text-Based Decomposition." ICLR. 2024.
>
> [2]Goh, Gabriel, et al. "Multimodal neurons in artificial neural networks." Distill 6.3 (2021): e30.

---

> > ### Comment · Reviewer_WdtL · 2025-08-05
> >
> > Thanks to the authors for their elaborate response to my questions. They have been fully addressed, and I have no further questions. I understand that it’s not possible at this time to provide comparisons with the other works I mentioned addressing the same issue. That said, based on W1, which remains to be the biggest weakness in my opinion, I will maintain my rating of borderline accept. The paper provides valuable insights into shortcomings of current CLIP models, but the proposed method is a temporary solution to a more fundamental problem.

---

> ### Author Response · Authors · 2025-08-04
> **Response to reviewer Wdtl**
>
> **A6**: We thank the reviewer for their patience and thoughtful question regarding the generalizability of our approach beyond the CLIP model. Unfortunately, we were unable to evaluate certain recent models due to practical constraints. For example, the **Vision Transformer Need Register [1] did not publicly release pretrained weights**, as noted in unresolved issues on their GitHub repository. Additionally, the **official code for the Perception Encoder became available only after the NeurIPS submission deadline**, and thus was not included in our evaluation.
>
> To validate the generalizability of our approach, we followed the suggestion of Reviewer mmh1 and evaluated our method on the SigLIP model. Specifically, we used the ViT-B variant as the SigLIP vision encoder and assessed our approach on five datasets that do not include a background category. The results are presented in the table below:
>
> | Method                | VOC20| City  | Context59 | ADE | Stuff | Avg |
> |----------------------|------|------|-------|------|------|-----------|
> | SigLIP | 48.0 | 20.5 | 18.6  | 11.5 | 12.1 | 22.1     |
> | SigLIP+ClearCLIP         | 5.7 | 2.9 | 1.7  | 0.6 | 1.6 | 2.3      |
> | **SigLIP+ours**             | **59.1** | **23.2** | **23.1** | **13.8** | **15.3** | **26.9**  |
>
> Similar to the CLIP model, the vanilla SigLIP model performs poorly on these datasets, achieving an average performance of only 22.1. Empirically, we observed that prior works such as ClearCLIP—which remove the feed-forward network (FFN) and residual connections and modify the attention mechanism in the final transformer layer—perform significantly worse than the vanilla SigLIP model. We hypothesize that this degradation stems from the architectural difference between CLIP and SigLIP: specifically, the replacement of the linear projector in CLIP with the AttentionPool projector in SigLIP. Modifying the final transformer layer, as done in ClearCLIP, may significantly alter the input distribution to the AttentionPool projector, negatively impacting performance. This warrants further investigation in future work. Accordingly, we adopt the vanilla SigLIP model as our baseline.
>
> Orthogonal to previous work that focuses exclusively on modifications to the final transformer layer, our approach is **grounded in a detailed analysis of spatial discriminability and semantic alignment across the token, head, and layer levels**. Rather than restricting changes to the final stage, we introduce improvements at earlier layers of the model, resulting in more effective and generalizable inference. As shown in the table, our method achieves a substantial improvement of **4.8** points in average segmentation performance, increasing from 22.1 to **26.9**.
>
> It is worth noting that, due to time constraints during the discussion period, we did not conduct extensive hyperparameter ablation or tuning. These results therefore demonstrate not only the **generalizability of our method across different CLIP variants** but also its **robustness to parameter selection**. We will include this discussion in the appendix in the revised version.
>
> Again, we thank the reviewer for raising this valuable suggestion. While we have conducted experiments on SigLIP to evaluate the generalizability of our method, we agree that further experiments on the Perception Encoder would be beneficial. Due to the limited duration of the discussion period, we were not able to include this evaluation in our initial response. However, we will follow the reviewer’s recommendation and include experiments on the Perception Encoder in the revised version of our submission.
>
> [1] Darcet, Timothée, et al. "Vision transformers need registers." arXiv preprint arXiv:2309.16588 (2023).

---

### Note · Authors · 2025-08-14

Dear Area Chair,

Thanks again for handling the reviews. For your convenience, we would like to provide a short summary of the discussion period from our point of view.  Overall, we are happy that all reviewers have now participated in discussion. Below, we summarize the results.

**Review Outcome**

Our submission initially received three Borderline Accepts (WdtL, tN1T and mmh1), one Borderline Reject (irCm) and one Reject(DEqc). After the rebuttal and discussion:

1. Reviewer **tN1T** increased the score, claiming that many concerns have been addressed.
2. Reviewer **WdtL** and **mmh1** maintain their scores, indicating that concerns are fully addressed and no further questions (WdtL) or our responses were acknowledged(mmh1)
3. Reviewer **DEqc** claims that our rebuttal has satisfactorily addressed the concerns.
4. Reviewer **irCm** had acknowledged our responses and posted a follow-up clarification question on the use of AUC scores, which we provided explanations and will reflect the changes.

**Summary of Responses to Major concerns**
1. **Hyperparameter Dependency (WdtL, tN1T, irCm, and DEqc)**:  We emphasize these hyperparameter are **generalizable** and **sensitive robustness** across various datasets based on our ablation study.

2. **Incomplete ablation (tN1T, irCm, DEqc)**: We clarify the table structure and provide additional ablation study for completeness.

3. **Lack of comparison with earlier layer outputs (WdtL and tN1T)**: We follow the reviewers’ suggestions and provide additional results of simply selecting an earlier layer output for segmentation.

4. **Generalization beyond CLIP model (WdtL and mmh1)**: we follow the suggestion from reviewer mmh1 and conduct additional experiments on SigLIP, validating the generalizability of our method.

5. **Novelty about abnormal tokens (tN1T and mmh1)**: we clarify the novelty of our discovery and insights, which are is acknowledged as interesting by reviewer WdtL and as rather important and transformational by reviewer tN1T and as valuable by reviewer mmh1.

**Commitments for the camera-ready version**

We will add new experiments and discussions to highlight the individual and complementary effects of each component in our approach, as well as its generalizability beyond the CLIP model. To improve readability, we will refine figures and table structures, incorporate additional citations, and provide clearer interpretations and clarifications of our methods to prevent misunderstanding.

Thank you

---

### Decision · Program_Chairs · 2025-09-17

**Decision:**

Reject

**Comment:**

Before the rebuttal this paper has received mixed ratings: 1 x Reject (`DEqc`),  1 x Borderline Reject (`irCm`), and 3 x Borderline Accept (`WdtL`, `tN1T`, `mmh1`).


All reviewers acknowledged the reported performance, the clarity and quality of the writing and the thorough evaluation.
The major criticisms for this work concern:
- the manual selection of the layers (potentially hindering the application to other backbones and datasets and the scalability of the approach);
- the high number of hyperparameters to tune (5-6);
- the practicality of the approach (using aggregate AUC scores for selecting heads in the Selective Head Enhancement module - SHE);
- the questionable rigor in the hyperparameter selection.


In the rebuttal the authors addressed in details the different concerns of the reviewers and reported additional results on SigLIP, showing performance boosts.


After the rebuttal and lively discussions between authors and reviewers, most reviewers are positive, with reviewer `DEqc` raising their rating to Borderline Accept and reviewer `tN1T` to Accept.
Reviewer `irCm` remains rather negative in their rating questioning the utilization of the usage of ground-truth labels from the training sets of the evaluated datasets for the selection of the heads.
This has sparked a discussion between reviewers.


Some main ideas from the discussion:
- The authors argue that the method is _training-free_ which is technically true, but they use AUC stats from 1000 images from the ground truth labels of the train set of VOC, Context, ADE20K and Stuff datasets. The approach is then compared with other training-free and unsupervised methods that do not train CLIP, do not leverage ground truth labels from these datasets, nor the actual label names themselves.

- The authors pointed out to the NACLIP to defend this practice of tuning hyperparameters on small sets from the evaluated datasets, however reviewer `WdtL` pointed out in discussions that this is not the case.

- Methods that train CLIP for semantic segmentation run the fine-tuning on datasets that are different from the ones used for evaluation. In any case, none of the prior methods had access to target labels to conduct open-vocabulary or zero-shot evaluation

- The concern of the reviewers is that given this distinction in practice that has already been missed by most reviewers in the first pass, other works may be tempted to follow the same approach as it can boost scores and undermine fair comparisons maintained in the benchmarks so far


The AC agrees with the arguments brought in the discussions. It is clear from the ablations (Table 5) that the SHE module contributes to performance improvements. However, it is unclear at this point how much impact the utilization of these ground-truth labels impact the final performance.
A few additional experiments would have helped clarifying how critical is the use of target data and labels:
- using a new different labeled dataset for the AUC computation, e.g., Mapillary, DUTS-TR, ImageNet-S
- using a different labeled dataset for each configuration: e.g., compute AUC on ADE20K and evaluated on Cityscapes


After extending the discussion with the Senior AC, the AC sides with reviewer `irCm` and recommends this submission for rejection.
The performance of the approach is high, but the evaluation protocol should be in line with the ones used by baseline methods and the introduction of a new evaluation protocol should be properly described and accompanied by a few baselines under the same protocol.
We encourage the authors to take into consideration the other useful advice from the reviewers towards improving this work for a potential resubmision.